# Cerebellar stimulation prevents Levodopa-induced dyskinesia in mice and normalizes activity in a motor network

Bérénice Coutant[1], Jimena Laura Frontera [1,3], Elodie Perrin[2,3], Adèle Combes[1,3], Thibault Tarpin[1], Fabien Menardy [1], Caroline Mailhes-Hamon[1], Sylvie Perez[2], Bertrand Degos[2], Laurent Venance [2], Clément Léna [1✉] & Daniela Popa [1✉]

Chronic Levodopa therapy, the gold-standard treatment for Parkinson's Disease (PD), leads to the emergence of involuntary movements, called levodopa-induced dyskinesia (LID). Cerebellar stimulation has been shown to decrease LID severity in PD patients. Here, in order to determine how cerebellar stimulation induces LID alleviation, we performed daily short trains of optogenetic stimulations of Purkinje cells (PC) in freely moving LID mice. We demonstrated that these stimulations are sufficient to suppress LID or even prevent their development. This symptomatic relief is accompanied by the normalization of aberrant neuronal discharge in the cerebellar nuclei, the motor cortex and the parafascicular thalamus. Inhibition of the cerebello-parafascicular pathway counteracted the beneficial effects of cerebellar stimulation. Moreover, cerebellar stimulation reversed plasticity in D1 striatal neurons and normalized the overexpression of FosB, a transcription factor causally linked to LID. These findings demonstrate LID alleviation and prevention by daily PC stimulations, which restore the function of a wide motor network, and may be valuable for LID treatment.

[1] Neurophysiology of Brain Circuits Team, Institut de biologie de l'Ecole normale supérieure (IBENS), Ecole normale supérieure, CNRS, INSERM, PSL Research University, 75005 Paris, France. [2] Center for Interdisciplinary Research in Biology (CIRB), College de France, CNRS, INSERM, Université PSL, Paris, France. [3]These authors contributed equally: Jimena Laura Frontera, Elodie Perrin, Adèle Combes. ✉email: clement.lena@bio.ens.psl.eu; daniela.popa@bio.ens.psl.eu

Motor symptoms of Parkinson's disease (PD) are caused by a progressive loss of dopaminergic neurons in the substantia nigra *pars compacta*, and of their dense projections to the striatum. The gold-standard symptomatic therapy for PD patients is Levodopa (L-DOPA). However, with disease progression and chronic exposure to L-DOPA, 50–80% of patients experience a wide range of motor levodopa-induced complications within 5-10 years of treatment[1] including debilitating abnormal involuntary movements, called levodopa-induced dyskinesia (LID)[2]. So far, very few therapeutic options are available to circumvent the advent of LID in the course of L-DOPA treatment. A better understanding of the brain networks controlling LID generation and expression is critical to the development of appropriate treatments.

LID-associated abnormalities have been consistently observed in the basal ganglia, the thalamus and the motor cortex in humans[3–6], primates[7–10] and rodents[11–15]. In line with these observations, interactions in this inter-connected motor network contribute to LID pathophysiology[16]. More recently, alleviation of LID in humans has been observed following stimulation of the cerebellum[5,17–20]. While a single short (1–2 min) session of repetitive transcranial magnetic (rTMS) continuous theta burst (cTBS) stimulation over the cerebellum only transiently reduced LID, the repetition of stimulation sessions over 2 weeks yielded to a reduction of peak-dose LID over weeks after the sessions[18,19]. This showed that cerebellar stimulation could reduce the expression of LID. The impact of this stimulation was observed in both the cerebellar hemispheres and the cerebellar nuclei of dyskinetic patients[17] and the authors suggested that their effect is mediated by the output cells of the cerebellar cortex, the Purkinje cells (PC), and propagated to downstream structures. However, how cerebellar stimulation modulates the activity of the cerebellum and other motor structures to alleviate LID and whether it acts on core mechanisms associated to LID are still unknown.

A first hypothesis would be that cerebellar stimulation corrects motor cortex dysfunction observed in dyskinesia. Indeed, dyskinetic patients present an increase in cerebral blood flow in the primary motor cortex[21], as well as abnormal synaptic plasticity[22]. Similarly, dyskinetic rats exhibit changes in gene expression[23] and an increased activity in about half of the neurons of the motor cortex[24]. In addition, subthalamic deep brain stimulation, which reduces PD symptoms and thus prevents the need of high L-DOPA dosage leading to LID, has been proposed to act via an effect on the motor cortex[25,26]. Likewise, cerebellar cTBS has been shown to exert a control on motor cortex plasticity[27]. Moreover, anodal direct current stimulation over the cerebellum, which is thought to increase the cerebello-cortical coupling[28], also led to a decrease in LID[29]. Therefore, the motor cortex could be the relay of cerebellar stimulation beneficial effects to the basal ganglia in the treatment of LID.

LID are also directly linked to abnormal molecular events taking place in striatal neurons[30,31]. Most notably, LID have been causally linked to changes in the expression of FosB, a transcription factor, and its truncated splice variant ΔFosB. Dyskinetic patients[32], primates[33,34] and rodents[35–39] show an overexpression of FosB/ΔFosB that strongly correlates with the severity of dyskinesia[38]. The upregulation of FosB/ΔFosB in striatal neurons of experimental animals is sufficient to trigger LID following acute administration of levodopa[7,40], and reciprocally, the inactivation of striatal FosB/ΔFosB reduces LID[34,41] establishing the causal contribution of this transcription factor with LID. In addition, LID are associated with strong changes in striatal synaptic plasticity[14,42]. These aberrant corticostriatal plasticities are indeed a feature shared with a number of other hyperkinetic movement disorders, suggesting that they participate to the pathological state[43,44]. Besides its cortical inputs, the

striatum receives massive inputs from the thalamus[45]. Considering all of this, an alternative hypothesis would be that the thalamo-striatal pathway might relay therapeutic activities as demonstrated by the reduction of LID following deep brain stimulation of the intralaminar thalamo-striatal CM-PF complex in PD patients[46] and dyskinetic rats[47]. The cerebellum indeed projects to the basal ganglia by way of the intralaminar thalamus[48–51], including the parafascicular nucleus[52–54] and may control corticostriatal plasticity[49]. Cerebellar stimulation could therefore restore striatal function in LID via the cerebello-thalamo-striatal disynaptic pathway.

To investigate the mechanisms underlying LID alleviation by cerebellar stimulation, we studied the effects of optogenetic PC stimulation on LID using L7-ChR2-YFP (Yellow Fluorescent Protein) mice[55] in combination with a well-known mouse model of LID[56]. We performed daily brief sessions of theta-rhythm optogenetic stimulations of PC in CrusII, the region associated with orolingual sensorimotor function of the cerebellum[57,58]. These stimulations did specifically suppress, or even prevent, if administered early enough, severe orolingual LID. These behavioral findings were paralleled with a normalization of the aberrant neuronal activities in the interposed nucleus, the oral primary motor cortex, and the parafascicular thalamus, indicating a wide-scale action of cerebellar stimulation on the motor system. The chemogenetic inactivation of the cerebello-parafascicular pathway counteracted the beneficial effects of cerebellar stimulation, suggesting that they are mediated via the cerebello-thalamo-striatal pathway. Indeed, cerebellar stimulation reversed the sign of corticostriatal plasticity by promoting long-term depression in D1-expressing neurons and normalized the striatal expression of FosB/ΔFosB indicating that cerebellar stimulation acts on the core mechanisms of LID genesis.

## Results

**Optogenetic stimulation of Purkinje cells in the orolingual region of the cerebellar hemisphere specifically suppress or prevent orolingual dyskinesia.** To study the effect of repeated daily sessions of optogenetic stimulations of PC on dyskinesia, we used a classical mouse model of LID[56,59]. LID were produced by repeated systemic injections of levodopa in mice that underwent severe dopaminergic depletion following 6-OHDA injection in the median forebrain bundle, which mainly projects to the dorsal striatum (Fig. 1a–c). All dopaminergically-depleted mice presented parkinsonian-like symptoms during the pre-levodopa phase (Fig. S1a) as observed by the classical test of the asymmetric use of the forepaw[60] (Fig. S1b and Table S1). As LID can be divided into different subtypes (oral, limb, and axial) (see "Methods") and the fractured somatotopy of the cerebellum has described a specific region in the cerebellar hemispheres, CrusII, hosting dense projections from the orolingual area[57], oral LID has been used as a powerful readout of the potential beneficial impacts of cerebellar stimulation centered on CrusII. All LID subtypes were scored on the fourth day of the protocol when animals received 6 mg/kg L-DOPA dosage (Fig. 1a). 6-OHDA-lesioned animals chronically treated with levodopa alone (condition "LID", $N = 19$) indeed exhibited severe oral, axial and limb dyskinesia, compared to control mice, also called SHAM animals (condition "SHAM", $N = 17$) (Fig. 1d–g). SHAM animals are unlesioned mice receiving daily dose of levodopa treatment. The dyskinesia score in LID mice peaked around 30–40 min after levodopa injection (Figs. S2b, S3b and S4b) as described in previous studies[61,62], consistent with LID severity following plasmatic levels of levodopa[56,63], hence referred to as peak-dose dyskinesia. These effects were observed during the 6 weeks of daily levodopa administration.

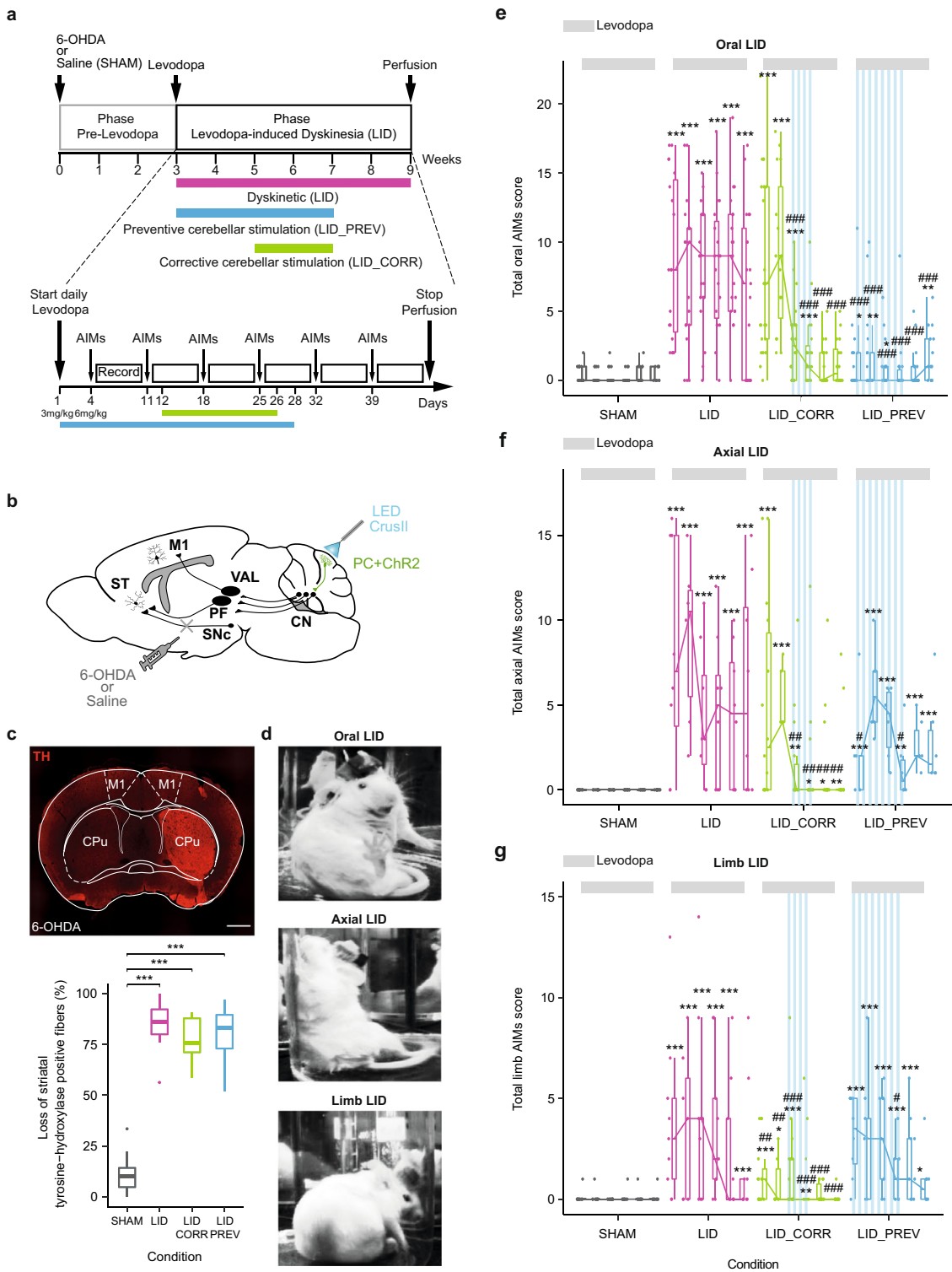

SHAM mice, exposed to chronic levodopa treatment, received either corrective or preventive optogenetic cerebellar stimulation or none. They exhibited only mild or no dyskinesia of any kind, neither before nor during cerebellar stimulation (Fig. S5), suggesting that cerebellar stimulation does not induce any kind of abnormal motor phenotype, and they were therefore pooled together for behavioral analysis.

We first attempted to reproduce the beneficial effects observed in dyskinetic patients[19] following two weeks of cerebellar stimulation in our mouse model of LID. To examine whether PC stimulation

efficiently reverses LID, brief trains of optogenetic stimulations at theta frequency[19,64] (see "Methods") were delivered daily in mice expressing ChR2-YFP specifically in PC[55,60] (Fig. 2b, c), at the peak of LID expression (30 min after injection), starting 2 weeks after the start of the levodopa treatment (6 mg/kg), thus after LID development ("corrective stimulation") (Fig. 1a). In 6-OHDA-lesioned mice exhibiting severe dyskinesia, 2 weeks of daily corrective PC stimulations on the cerebellar orolingual region CrusII significantly reduced oral dyskinesia (condition "LID_-CORR", $N = 24$, Fig. 1e, S2b and Table S2). This beneficial effect

**Fig. 1 Optogenetic stimulation of Purkinje cells in the orolingual region of the cerebellar hemisphere specifically suppresses or prevents orolingual dyskinesia. a** Experimental timeline. Dyskinetic mice (LID, magenta): 6 weeks of levodopa treatment. Preventive mice (LID_PREV, blue): 6 weeks of levodopa treatment + 4 weeks of cerebellar stimulation. Corrective mice (LID_CORR, green): 6 weeks of levodopa treatment + 2 weeks of cerebellar stimulation. AIMs corresponds to the evaluation of Abnormal Involuntary Movements (LID). **b** Sagittal schematic of a mouse brain showing cerebello-thalamo-cortical and -striatal pathways, ChR2-YFP in Purkinje cells (PC + ChR2, green), and injection site of 6-OHDA or saline. M1: Primary motor cortex, ST: striatum, VAL: Ventroanterior-ventrolateral complex of the thalamus, PF: Parafascicular nucleus of the thalamus, SNc: Substantia nigra *pars compacta*, CN: cerebellar nuclei, CrusII: Crus2 of the ansiform lobule. **c** *Upper panel*: Coronal section from a mouse unilaterally-lesioned with 6-OHDA stained with anti-tyrosine hydroxylase (TH). Scale bar: 0.5 mm. *Bottom panel*: Boxplot showing the average loss of striatal TH-positive fibers (%) between the lesioned and the intact striatum in SHAM (gray, $N = 17$), LID (magenta, $N = 13$), LID_CORR (green, $N = 10$), and LID_PREV (blue, $N = 17$). **d** Examples of orolingual (top), axial (middle), and limb (bottom) levodopa-induced dyskinesia in dyskinetic mice. **e** Boxplot showing the sum of oral LID scores across the 6 weeks of levodopa treatment (6 mg/kg) (light gray bar) for SHAM ($N = 18$), LID ($N = 19$); LID_CORR ($N = 17$); LID_PREV ($N = 24$). **f** Boxplot showing the sum of axial LID across the 6 weeks of levodopa treatment (6 mg/kg) (light gray bar) for SHAM ($N = 12$), LID ($N = 8$), LID_CORR ($N = 14$), LID_PREV ($N = 6$). **g** Boxplot showing the sum of limb LID scores across the 6 weeks of levodopa treatment (6 mg/kg) (light gray bar) for SHAM ($N = 14$), LID ($N = 9$), LID_CORR ($N = 15$), LID_PREV ($N = 9$). Boxplots represents the lower and the upper quartiles as well as the median of LID score. Each boxplot corresponds to 1 week. Vertical lines represent the median ± the standard deviation. Isolated points represent outliers of the distribution. Stripped blue lines: weeks of theta-burst PC stimulation. Kruskal-Wallis test with pairwise two-sided Wilcoxon test and Benjamini & Hochberg correction. ***$p < 0.001$; **$p < 0.01$; *$p < 0.05$; * compared to SHAM; # to LID. Source data are provided as a Source Data file. See also Table S1.

outlasted the end of cerebellar stimulation for 2 weeks (weeks 8 and 9, Fig. 1e, S2b and Table S2) and the dyskinesia scores were then similar to those of the SHAM group (Table S2). Furthermore, after corrective PC stimulations over orolingual CrusII, the reduction in oral LID was more pronounced than in axial and limb dyskinesia (Fig. 1f, g, S2b and S4b). We were therefore able to reproduce the beneficial effects of cerebellar stimulation observed in dyskinetic patients on mice, confirming the usefulness of our model to study the modulations underlying cerebellar stimulation in the alleviation of LID.

We next wondered whether cerebellar stimulation could prevent the development of LID when administrated at the beginning of the levodopa treatment. To answer this question, another group of 6-OHDA-lesioned mice received daily cerebellar stimulation starting from the first day of levodopa administration (3 mg/kg) ("preventive" stimulation) i.e before the development of dyskinesia (condition "LID_PREV", $N = 18$). Preventive mice greatly differed from corrective animals as they did not develop LID before receiving cerebellar stimulation therefore, the initial state of the motor circuits is profoundly different when cerebellar stimulation started. Remarkably, the preventive group exhibited only mild to no orolingual dyskinesia, contrarily to unstimulated LID animals (Fig. 1e, S1b and Table S1). In conclusion, 2 weeks of daily cerebellar stimulation led to a dramatic decrease of LID expression that outlasted the stimulation, while stimulation starting concomitantly with levodopa administration prevented LID development, up to two weeks after the stimulation ended. Therefore, these results indicate a strong and long-lasting suppressive effect of peak-dose dyskinesia by cerebellar PC stimulation.

Previous studies in animal models of LID addressed exclusively "peak-dose" dyskinesia[65]. Yet, mild dyskinesia also occurred outside the 2 hours-window following levodopa uptake in PD patients at the trough of blood levodopa concentration ("off-period" dyskinesia) or during the rising and falling phase of blood levodopa concentrations (diphasic dyskinesia) as review in[66]. Therefore, analysis of dyskinesia observed 20 min before levodopa injection revealed that chronic PC stimulations on CrusII also suppressed or prevented oral "off-period" dyskinesia (Fig. S1c and Table S2).

In conclusion, daily sessions of opto-stimulation of PC in CrusII, which corresponds to the orolingual region of the cerebellar cortex, are sufficient to obtain a significant decrease of oral LID. These results bear resemblance with those obtained in PD patients in whom rTMS targeting posterior cerebellum improved LID scores[18,19] but show a stronger effect than in humans where the severity of dyskinesia was only reduced at the peak effect of levodopa. If administrated before LID development, cerebellar stimulation can even prevent LID appearance in preventive mice. Therefore, preventive animals represent a stronger and more interesting condition for the study of the mechanisms underlying LID alleviation and treatment. The rest of study will thus focus on this condition.

**Purkinje cell stimulation over CrusII prevents the depressed activity of the interposed nucleus observed in LID mice.** To test whether levodopa treatment results in changes of activity in the cerebellum, we chronically recorded neurons in the three cerebellar nuclei (CN): the interposed (IN), the dentate (DN), and the fastigial nuclei (FN) (Fig. 2b and S6b). Neuronal activity was recorded both before and after levodopa administration in freely moving 6-OHDA-lesioned and SHAM mice, for a total of 9 weeks (Fig. 2a, b and S6a–c). Recordings of CN neurons in three mice during theta-burst stimulation protocol revealed that cells in the three CN were strongly inhibited by PC stimulations (hence likely receiving inputs from the stimulated area) and exhibited an increased firing rate relative to the baseline activity when stimulation stopped (Fig. 2c). Because of the high variability of the firing rate in CN[67] and in the different conditions before the start of the levodopa treatment, we decided to focus on the evolution of the global activity throughout the protocol and only make intra-group comparisons. We observed that levodopa decreased the global activity of IN and DN, but not FN, in LID animals (Fig. 2d, S6d–f, Tables S4 and S5). We verified that this did not reflect changes in motor activity as no differences were observed in CN activity during levodopa treatment in SHAM mice, whether the animals were moving or not[68,69] (Supp. Text, Fig. S6 and S7). Altogether, these results constitute the first evidence of a dysregulated activity of the output nuclei of the cerebellum in LID.

Moreover, in preventive animals, receiving 4 weeks of PC stimulation at the beginning of the levodopa treatment (LID_PREV), we found that the depressed activity in IN, induced by levodopa treatment in LID mice, was prevented following preventive PC stimulation (Fig. 2d and Tables S3). However, no significant prevention of the decrease in global activity in DN was observed following PC stimulation suggesting that this nucleus is less recruited in LID alleviation following preventive cerebellar stimulation (Fig. S6d–f and Tables S3). Taken together, these data show that repeated sessions of cerebellar stimulation affect the aberrant activity observed under levodopa treatment in IN, which suggests a prominent role of this structure in the normalization of dyskinesia.

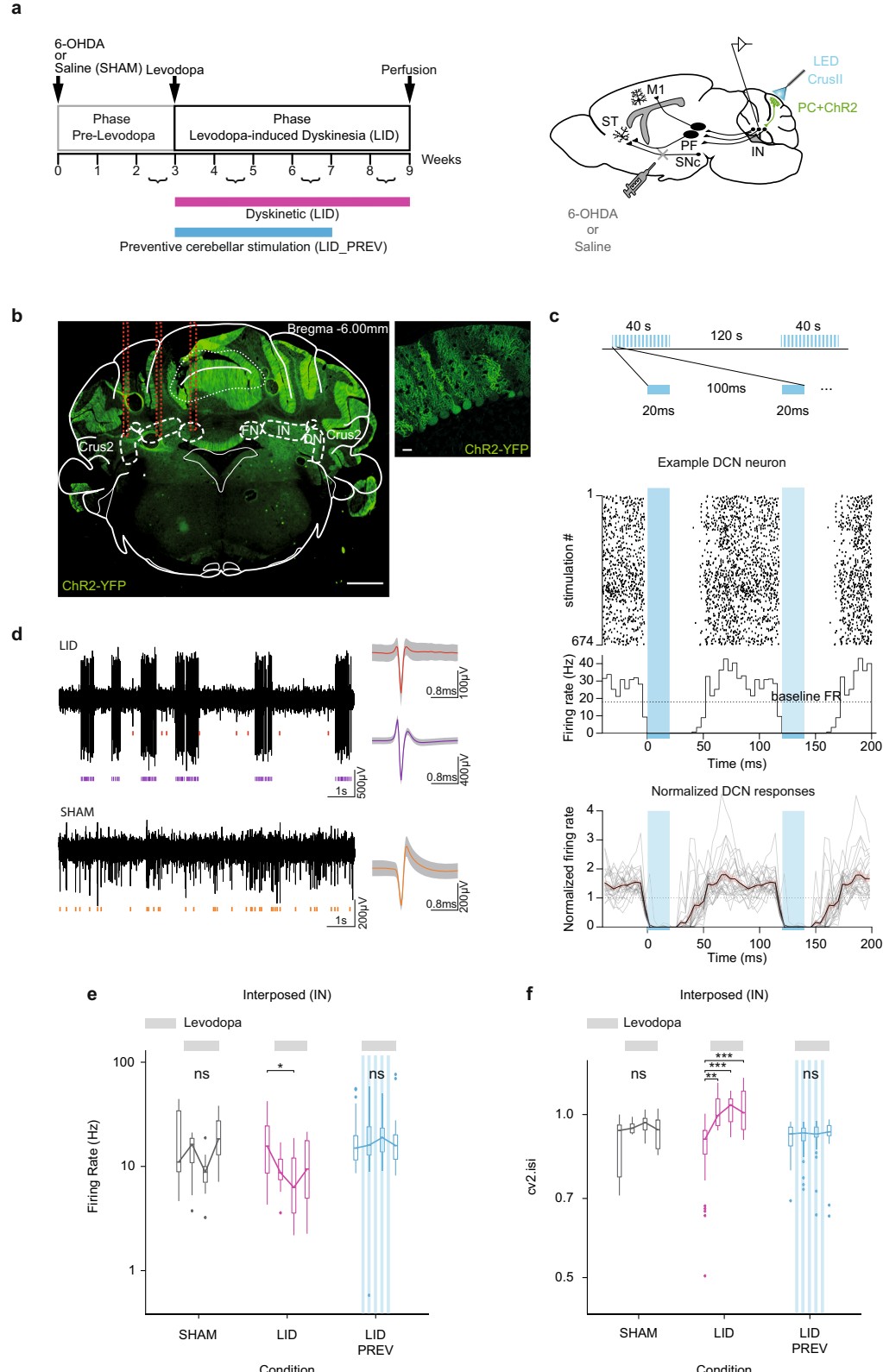

**Purkinje cell stimulation over CrusII prevents the erratic activity of the interposed nucleus observed in LID mice**. The irregularity of neural discharge in the cerebellum is detrimental to motor control[70] and has been observed in rapid-onset dystonia-Parkinsonism[71,72] and tremor[73]. The average normalized difference of successive interspike intervals (cv2.isi) is a measure of irregularity of directly adjacent interspike intervals, and therefore higher cv2.isi value indicates a more irregular cell activity[74]. LID mice exhibited a higher cv2.isi in the IN, DN and FN during the entire period of levodopa treatment (Fig. 2e, S6e–g, Tables S3 and S4). The higher values of cv2.isi did not simply reflect increased bursting as the burst rate of LID mice decreased during levodopa treatment in the IN during locomotor activity (Supp. Text, Fig. S9a). Therefore, these results showed a more erratic and

**Fig. 2 Purkinje cell stimulation normalizes firing rate and regularizes pattern of activity in the interposed nucleus. a** *Left*: Experimental timeline. *Right*: Schematic of electrode implantation in the interposed nucleus (IN), ChR2-YFP expression in Purkinje cells (PC + ChR2, green) and injection site of 6-OHDA or saline. ST: Striatum; SNc: substantia nigra *pars compacta*; M1: Primary motor cortex; PF: Parafascicular nucleus of the thalamus. **b** *Left*: Coronal section from L7-ChR2-YFP mouse. Red lines: electrode's trajectory. Dotted white lines: IN, dentate (DN), and fastigial (FN) nuclei. Scale bar: 0.5 mm. Crus2: Crus2 of the ansiform lobule. *Right*: PC expressing YFP. Scale bar: 20 µm. **c** *Top*: Theta-burst protocol. *Middle*: Raster plot of a cerebellar nuclei (CN) neuron for each stimulation. Dotted line: Basal firing rate (FR) before the onset of stimulation. Blue box: Time of optogenetic stimulation. *Bottom*: Summary of CN firing profiles ($n = 27$; $N = 3$) exhibiting a strong inhibition (>90%) during PC stimulation. The firing rate of each unit was normalized to its baseline. Shaded lines: mean ± the standard deviation (std). **d** Examples of raw traces recorded in IN in LID (*top*) and SHAM (*bottom*) and their associated spike shape. **e** Firing rate (Hz) across 9 weeks in IN. Boxplots show the median (horizontal bars) over 4 categories of weeks. First boxplot: 2nd and 3rd weeks; second boxplot: 4th and 5th weeks when levodopa begins; third boxplot: 6th and 7th weeks; last boxplot: 8th and 9th weeks when stimulation stopped. **f** Coefficient of variation 2 (cv2.isi) across 9 weeks in IN. Same order of boxplot as panel **e**. Gray = SHAM ($N = 5$); Magenta = LID ($N = 3$); Blue = LID_PREV ($N = 4$). Light gray lines: 6 weeks of levodopa treatment (3 boxplots; 6 mg/kg). Stripped blue lines: weeks of theta-burst stimulation. Boxplot represents the lower and the upper quartiles as well as the median rate (horizontal bars). Vertical lines represent the median ± std. Isolated points represent outliers of the distribution. Welch Anova with two-sided Games Howell post-hoc test and one-way Anova's with two-sided Tukey post-hoc test based on Levene test. \*\*\**p* < 0.001; \*\**p* < 0.01; \**p* < 0.05; ns: *p* > 0.5. Source data are provided as a Source Data file. See also Tables S4–S6.

---

irregular pattern in CN neurons in LID mice during levodopa treatment.

In preventive mice, receiving 4 weeks of PC stimulation (LID_PREV), we found that the increased cv2.isi in IN was prevented by PC stimulation (Fig. 2e, Tables S3 and S4). However, increased cv2.isi was still present in DN and FN (Fig. S6e–g, Tables S4, S6), reinforcing the potential crucial role of the IN in LID alleviation. Overall, these electrophysiological data suggest that dyskinesia-related abnormal activity is conveyed to the CN, especially to IN, leading to aberrant firing rate and firing patterns, which are reversed by chronic PC stimulation. Finally, these experiments suggest a tighter association between the changes in IN activity and the alleviation of the pathological phenotype.

**Chronic levodopa treatment increases the activity of the oral motor cortex and decreases the activity in the parafascicular thalamic nucleus of dyskinetic mice.** Since LID strongly involve the forebrain motor circuits[16], and cerebellar nuclei have multiple ascending projections toward these circuits[75,76], we next investigated the impact of cerebellar stimulation on the thalamus and motor cortex. Changes in motor cortex activity, intralaminar nuclei of the thalamus, including the parafascicular nucleus (PF), and the ventroanterior-ventrolateral (VAL) complex of the thalamus have been observed in dyskinetic patients[21,22,46] and animals[23,24,47]. Moreover, cerebello-cortical loops[77,78] and parafascicular projections to the striatum and the cerebral cortex[79] are topographically organized. Therefore, to examine the impact of both levodopa treatment and cerebellar stimulation on the thalamus and motor cortex over time, we chronically recorded neurons in the oral region of M1, in PF and the cerebellar-recipient area of VAL, during 5 weeks of chronic levodopa treatment (Fig. 3a, b and S10a, b). Chronic recordings allowed us to study the evolution of the global activity of these structures across levodopa treatment as well as the long-term effects of cerebellar stimulation in preventive mice.

The activity in M1 and PF varied slightly over the course of levodopa treatment in SHAM animals, whereas LID mice exhibited a significant increase of the firing rate in M1 (Fig. 3c, Tables S8, S9) and a significant decrease in the firing rate in PF after levodopa administration (Fig. 3d, Tables S8 and S9). No modulation of the global activity in the cerebellar region of VAL was observed following levodopa treatment (Fig. S10d, Tables S8, S9), suggesting that VAL is less recruited for LID expression. Altogether, these results confirm the presence of functional alterations in PF and oral M1 in dyskinesia.

**Purkinje cell stimulation over CrusII prevents both the abnormal increase in M1 and the aberrant decrease in PF.** We then investigated whether the abnormal activities observed in M1 and PF could also be prevented by chronic PC stimulations as observed in the IN.

In preventive animals, receiving 4 weeks of cerebellar stimulation (LID_PREV), both the aberrant increased activity in oral M1 (Fig. 3c, Tables S5 and S6) and the abnormal decreased activity in PF observed in LID were prevented (Fig. 3d, Tables S8 and S9), and remained normal even after the end of the stimulations. As for LID mice, no significant modulation of the activity in the cerebellar-recipient area of VAL was observed in preventive mice following cerebellar stimulation, further suggesting that both levodopa and repeated sessions of cerebellar stimulation have less impact on this structure (Fig. S10d, Tables S8 and S9).

Taken together, these results suggest that the effects induced by cerebellar stimulation prevents aberrant activities in both oral M1 and PF, but not in VAL, by reversing the changes in firing rate associated with dyskinesia.

**Cerebellar nuclei send massive projections to PF with few collaterals to VAL.** Because CN and PF, but not VAL, showed similar modulations of their firing rate in both dyskinetic and preventive mice following levodopa treatment and cerebellar stimulation respectively, we examined how the IN, DN, and FN are projecting to PF and whether these projections form collaterals to VAL. For this purpose, we used retrograde viral tracing to determine the distribution of CN neurons projecting to PF (Fig. 4a). Quantification of retrograde labeled neurons from PF showed 42.1 ± 9.0 % of projecting neurons in IN, 41.4 ± 2.6 % of neurons in DN, and 16.5 ± 1.7 % of neurons in FN (N = 6, Fig. 4b–d), suggesting that IN and DN might have an important contribution in the cerebellar control of PF activity. Moreover, PF-projecting CN neurons were localized along the entire antero-posterior axis in the three nuclei (Fig. 4b, c, e).

To confirm the presence of CN synaptic terminals in PF neurons, we localized the synaptic terminals of CN neurons projecting to PF using Cre-dependent viral expression of the presynaptic marker synaptophysin (SynP)-GFP (Green Fluorescent Protein) in combination with the expression of Cre-recombinase obtained by retrograde viral injections in PF (Fig. 4f). Large amounts of CN terminals expressing SynP-GFP were found in PF (Fig. 4g, h) with very little collaterals to VAL (Fig. 4i, j). Overall, these results confirm the presence of cerebellar projections to PF and demonstrate that they originate from a population distinct from the one projecting to VAL, confirming

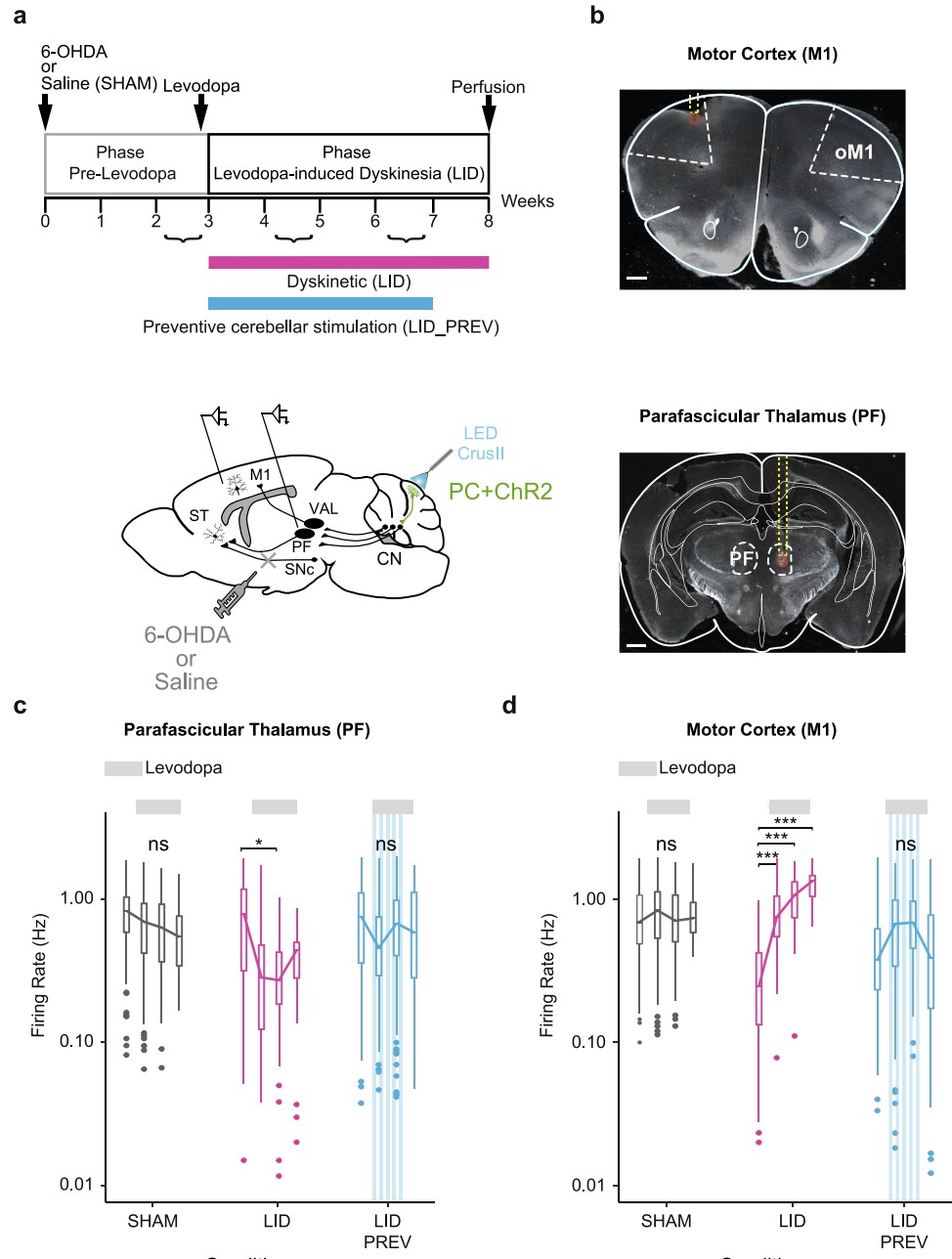

**Fig. 3 Aberrant activities in the motor cortex and the parafascicular nucleus of the thalamus in dyskinesia are restored by Purkinje cell stimulation. a** *Top*: Experimental timeline. *Bottom*: Schematic of electrode implantation in the primary motor cortex (M1) and the parafascicular nucleus of the thalamus (PF), ChR2-YFP expression in the Purkinje cells (PC + ChR2, green), and injection site of 6-OHDA or saline. ST: Striatum; SNc: substantia nigra *pars compacta*; CN: cerebellar nuclei: VAL: Ventroanterior-ventrolateral complex of the thalamus. **b** Top: Coronal section from L7-ChR2-YFP mouse showing the electrode's trajectory (dotted yellow line) and the lesion site (red circle) in layer 5 of oM1. Scale bar: 0.5 mm. Bottom: Coronal section from L7-ChR2-YFP mouse showing the electrode's trajectory (dotted yellow line) and the lesion site (red circle) in PF. Scale bar: 0.5 mm. **c** Firing rate (Hz) across 9 weeks in M1. Boxplots show the median rate (horizontal bars), over 4 categories of weeks. First boxplot: 2nd and 3rd week of the protocol, second boxplot: 4th and 5th weeks when levodopa begins, third boxplot: 6th and 7th weeks, last boxplot: 8th of the protocol when stimulation stopped. Gray = SHAM (*N* = 5); Magenta = LID (*N* = 4); Blue = LID_PREV (*N* = 8). Light gray lines: 6 weeks of levodopa treatment (3 boxplots; 6 mg/kg). Stripped blue lines: weeks of theta-burst stimulation. **d** Firing rate (Hz) across 9 weeks in PF. Same order of boxplot as panel **c** Gray = SHAM (*N* = 5); Magenta = LID (*N* = 4); Blue = LID_PREV (*N* = 8). Light gray lines: 6 weeks of levodopa treatment (3 boxplots; 6 mg/kg). Stripped blue lines: weeks of theta-burst stimulation. Boxplot represents the lower and the upper quartiles as well as the median of the firing rate (horizontal bars). Vertical lines represent the median ± std. Isolated points represent outliers of the distribution. One-way Anova with two-sided Tukey HSD post-hoc test. ***$p < 0.001$; **$p < 0.01$; *$p < 0.05$; ns: $p > 0.5$. Source data are provided as a Source Data file. See also Tables S8 and S9.

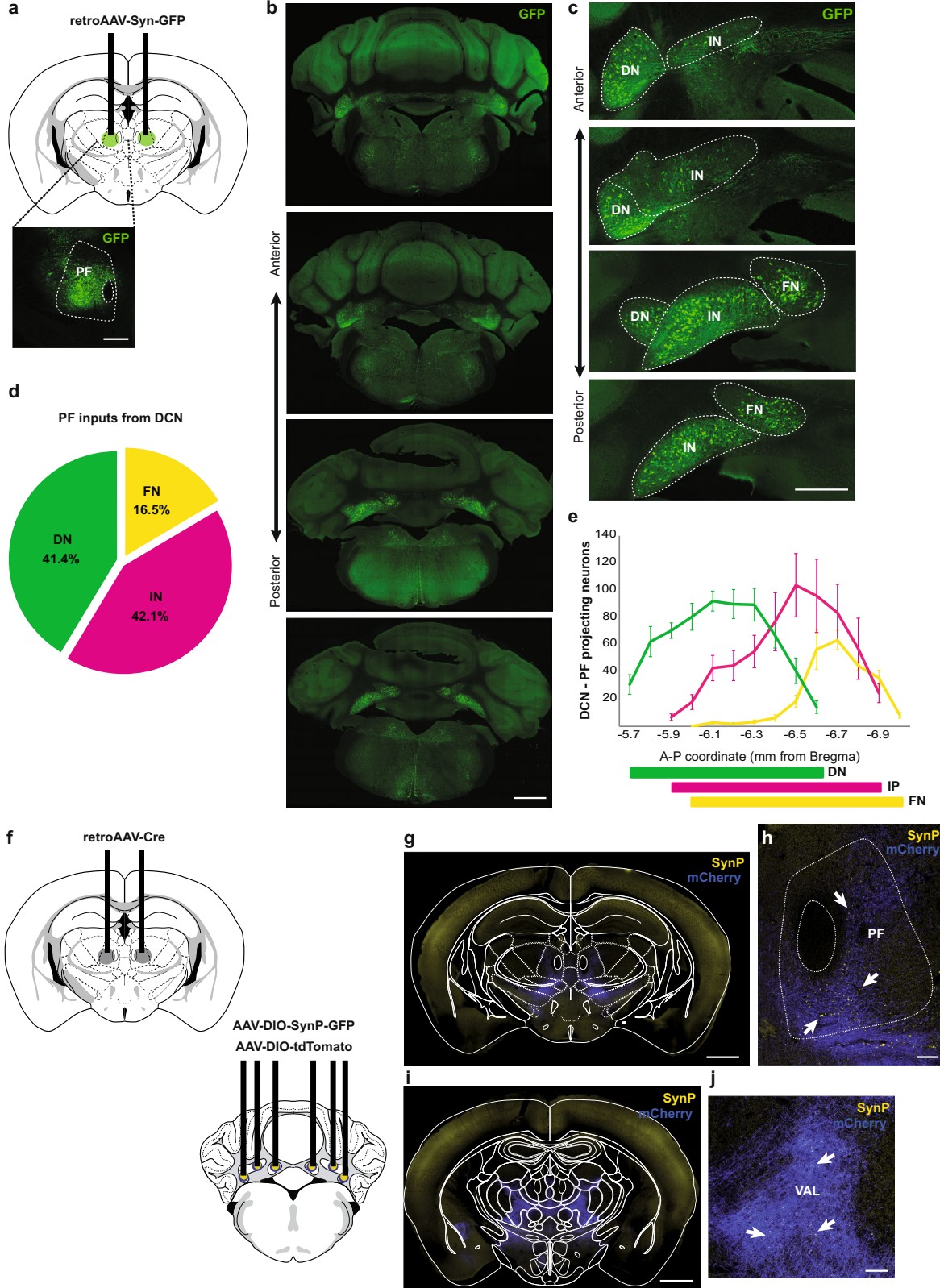

that cerebellar stimulation might modulate similarly the activity of the CN and PF, without altering the activity of VAL.

**CN-PF pathway inhibition counteracts the beneficial effects of Purkinje cell stimulation on oral dyskinesia in preventive mice.** As both CN and PF present similar modulations following PC

stimulation that could be transmitted from CN projections to PF, we examined the specific involvement of the CN-PF pathway in the beneficial impacts of cerebellar stimulation on LID. To do so, we examined the effects of transient inactivation of the specific projections from CN to PF during repeated sessions of PC opto-stimulations. For this purpose, we expressed inhibitory hM4Di DREADD receptors in CN neurons that target PF, by injecting

**Fig. 4 CN monosynaptic inputs to PF and collaterals. a** Retrograde labeling strategy by viral injection of retrograde AAV-syn-GFP in the parafascicular nucleus of the thalamus (PF). Inset: Post-mortem histology showing the injection site of retrograde AAV-syn-GFP in PF. Green = GFP. Scale bar: 0.5 mm. **b** Anterior to posterior cerebellar sections showing retrograde labeled neurons in the three cerebellar nuclei (CN): dentate (DN), interposed (IN), and fastigial (FN). Scale bar: 1 mm. **c** High-magnification on CN from **b**. Scale bar: 0.5 mm. **d** Quantification of the cell fraction (%) of retrograde labeled DN ($N = 6$), IN ($N = 6$) and FN ($N = 6$) neurons projecting to PF. Data are presented as mean ± sem. **e** Distribution of retrograde labeled CN neurons projecting to PF in each nucleus. **f** Tracing of axon collaterals from CN-PF projecting neurons by expression of retrograde AAV-Cre in PF, AAV-DIO-SynP-GFP and AAV-DIO-tdTomato in IN, FN, and DN. **g** Posterior thalamic section showing CN neurons projecting to PF expressing Cre-dependent Synaptophysin (SynP)-GFP and tdTomato. Scale bar: 1 mm. **h** Zoom-in of PF section exhibiting synaptic boutons (arrows) in CN inputs. Scale bar: 100 µm. **i, j** Anterior thalamic section showing axon collaterals within the thalamus. Scale bar: 1 mm. **j** Zoom-in from **i** of VAL section exhibiting some synaptic boutons (arrows) from CN-PF axon collaterals. Scale bar: 1 mm. Source data are provided as a Source Data file. Anatomical schemes were taken from Franklin and Paxinos Brain Atlas, 3rd edition.

retrograde CAV2-Cre-GFP in PF in complementation with a Cre-dependent AAV-DIO-hM4Di-mCherry in CN (Fig. 5a–d) in mice receiving 4 weeks of PC stimulation. This viral complementation has been shown to decrease the firing rate of CN neurons up to 30%[80]. Moreover, we showed that 24.18% ± 1.70% of CN neurons in LID_PREV saline and 22.72% ± 1.18% of CN neurons in LID_PREV CNO were expressing hM4Di DREADD (Fig. 5e), suggesting that this viral complementation would allow significant inhibition of these specific projections.

As oral LID peaked around 30 min after levodopa injection, we scored oral LID severity in preventive mice at this time point, and after cerebellar stimulation, to highlight the difference of effects between chemogenetic inhibition and optogenetic activation. Mice injected with inhibitory DREADD, stimulated for 4 weeks and receiving daily CNO injections before cerebellar stimulation, presented significantly more severe oral dyskinesia than saline preventive mice which were also injected with inhibitory DREADD, stimulated for 4 weeks but only received daily saline injections ($p < 0.001$; Fig. 5e). These results indicate that CN to PF inputs are required in the preventive effect of PC stimulation.

**Purkinje cell stimulation alters corticostriatal transmission and plasticity in brain slices.** As PF massively projects to the striatum[79,81,82], it may relay a cerebellar control over corticostriatal synaptic plasticity[49]. Indeed, alterations in corticostriatal plasticity are found in hyperkinetic disorders such as LID[14,43]. LID have notably been associated with an excessive corticostriatal long-term potentiation (LTP) in the direct pathway medium spiny neurons (MSN) without prominent change in the indirect pathway, resulting in an increased motor activity[43]. Therefore, the beneficial impacts of cerebellar stimulation on LID alleviation might act on the aberrant sign of corticostriatal plasticity within the striatum and might explain the observed long-term effects. Direct and indirect pathway MSNs express different dopaminergic receptors, the dopamine receptor subtype-1 (D1R) or subtype-2 (D2R) respectively for the direct and indirect pathways[83]. Therefore, we examined ex vivo the corticostriatal synaptic plasticity in MSNs of the dorsolateral striatum belonging either to the direct or indirect striatal pathways using brain slices from L7-ChR2xDrd2-GFP mice subjected to 4 conditions: SHAM, SHAM_PREV, LID or LID_PREV (Fig. 6a). We previously reported that using a spike-timing dependent plasticity (STDP) paradigm, paired pre-synaptic activations preceded by post-synaptic activations induced LTP in both direct and indirect pathway MSN[84,85]. LTP was induced in MSNs of SHAM, SHAM_PREV, LID or LID_PREV mice except in direct pathway MSNs issued from LID_PREV mice, where a clear long-term depression (LTD) was found (Fig. 6b–e and Table S13). Moreover, in direct pathway MSNs, LTP induced in LID mice exhibited greater magnitude ($p < 0.001$) than in SHAM mice, whereas preventive PC stimulation either reduced LTP in SHAM mice

($p < 0.001$) or even reversed LTP into LTD in LID mice. The magnitude of corticostriatal LTP induced in indirect pathway MSNs did not show significant variation in SHAM and LID mice with or without preventive PC stimulation (ANOVA: F = 0.3044, 38 df, $p = 0.822$).

Therefore, we found that cerebellar stimulation reverses striatal pathological LTP into LTD in direct pathway neurons, an effect which may then prevent the consolidation of an abnormal motor activity in the direct pathway and help reinstating normal motor functions.

**Purkinje cell stimulation normalizes the expression of the dyskinetic marker FosB/ΔFosB in the dorsolateral striatum.** The expression of the transcription factor FosB/ΔFosB, from the immediate early gene *fosb*, has been used as a marker of dyskinesia[38]. Alterations in its expression within the dorsolateral striatum affects LID, as both its inactivation[41] and its upregulation[7,40] can respectively reduce and increase the severity of dyskinesia. We then examined whether PC stimulation, leading to long-term LID alleviation, also normalizes the expression of the dyskinetic marker FosB/ΔFosB in the dorsolateral striatum (Fig. 7a).

We compared FosB/ΔFosB expression in the dorsolateral striatum ipsilateral to the lesion with the dorsolateral striatum contralateral to the lesion, in a subset of animals in the different conditions. As expected, neither levodopa nor cerebellar stimulation impacted the basal expression of FosB/ΔFosB in SHAM animals and no asymmetry was found between the two striatum (SHAM, $N = 10$, 99.6 ± 1.3 %, Fig. 7b and Table S14). As previously demonstrated in other studies, LID mice presented an overexpression of FosB/ΔFosB in the dorsolateral striatum ipsilateral to the lesion (LID, $N = 10$, 128.9 ± 5.0 %, Fig. 7b and Table S14), significantly different from SHAM. However, no significant asymmetry of striatal FosB/ΔFosB expression was found in mice receiving 4 weeks of cerebellar stimulation (LID_PREV, $N = 10$, 94.2 ± 3.4 %, Fig. 7b and Table S14), which did not significantly differ from SHAM mice, suggesting that cerebellar stimulation prevents the overexpression of FosB/ΔFosB in preventive mice and restores its expression to control levels. Moreover, striatal FosB/ΔFosB expression in preventive animals was significantly different from the one observed in LID mice. Thus, our results suggest that PC stimulation is able to normalize the expression of striatal FosB/ΔFosB, which is accompanied by a long-lasting anti-dyskinetic effect.

In conclusion, repeated sessions of PC stimulation over Crus II can both normalize the aberrant activity of major motor structures involved in LID, including CN, PF, and M1, but also the overexpression of the dyskinetic marker FosB/ΔFosB in the dorsolateral striatum, which is tightly linked to the development of LID. These effects were associated with the advent of LTD in the striatal direct pathway MSNs, which may prevent the

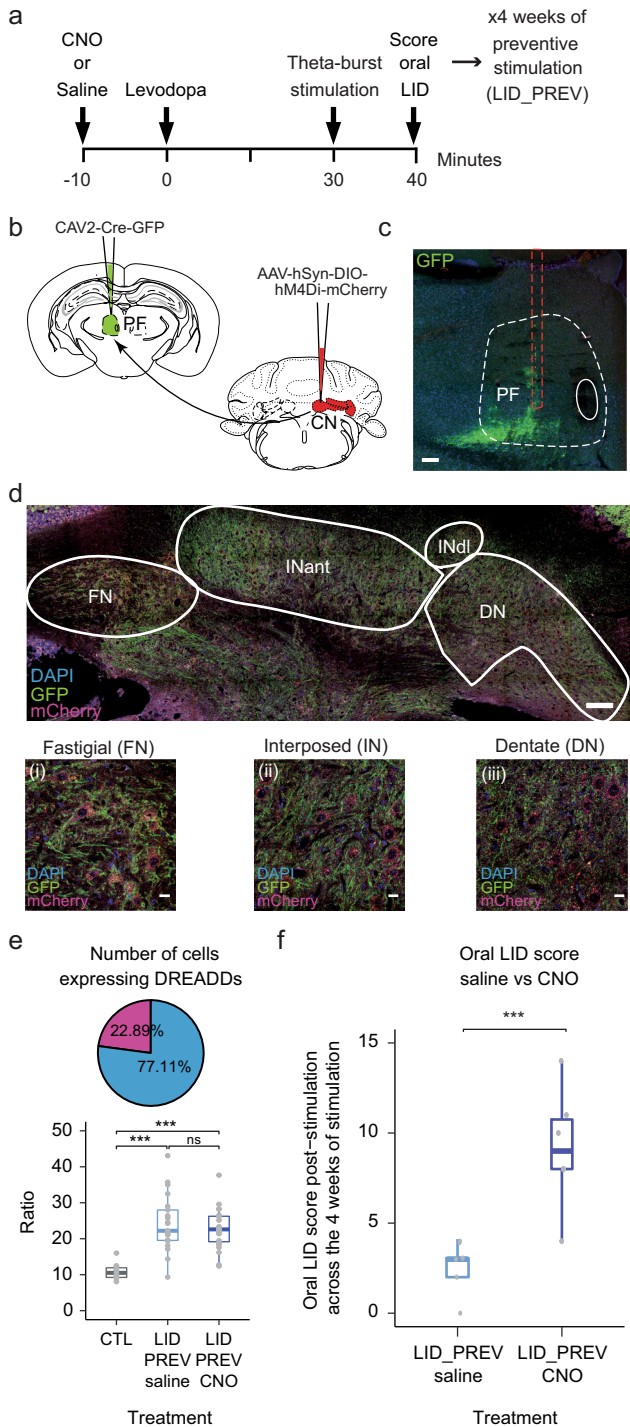

**Fig. 5 CN to PF pathway inhibition counteracts the beneficial effects of Purkinje cell stimulation on the severity of oral dyskinesia in preventive mice. a** Experimental timeline. **b** Schematic of mouse coronal sections showing the injection site of the retrograde CAV2-Cre-GFP in the parafascicular nucleus (PF, green) ipsilateral-to-the-lesion (*top, left*) and the injection site of the anterograde Cre-dependent pAAV-hSyn-DIO-hM4Di-mCherry in the three cerebellar nuclei (CN, red) contralateral-to-the-lesion. **c** Coronal section showing the injection site of retrograde CAV2-Cre-GFP in PF. Red dotted lines: needle's trajectory. White dotted lines: limits of PF. Scale bar: 100 μm. **d** Coronal section showing the expression of anterograde hM4Di-mCherry in neurons (red) of CN. FN: fastigial nucleus; INant: interposed nucleus, anterior part; InDL: interposed nucleus, dorsolateral hump; Lat: lateral nucleus. Blue = DAPI; green = GFP and YFP; Red = mCherry. Scale bar: 100 μm. Insert: Postmortem histology showing hM4Di-mCherry-expressing neurons in (i) FN, (ii) IN, and (iii) DN. Scale bars: 20 μm. **e** *Top*. Pie chart representing the proportions of neurons in CN expressing hM4Di-mCherry (in magenta) and projecting to PF compared to other types of cells (in blue). *Bottom*. Boxplots showing the average distribution (in %) of the neurons expressing hM4Di-mCherry over the total number of cells (Ratio) in control animals (non-virally injected animals, in gray) and preventive animals either treated with saline (light blue) or CNO (dark blue). Gray dots represent individual data points present in the distribution **f** Boxplots showing the average scored of oral LID severity at the point time corresponding to 40 min after levodopa injection, 50 min after CNO injection and after theta-burst stimulations. Averaged score comprises the 4 weeks of cerebellar stimulation in two groups: preventive animals receiving saline (LID_PREV saline, *N* = 5), dark blue represents preventive animals receiving CNO (LID_PREV CNO, *N* = 6). Boxplots represent the median score (horizontal bars), the lower and the upper quartiles. Vertical lines represent median ± std. One-way Anova with two-sided Tukey HSD post-hoc test was used for 5e and non-parametric Kruskal-Wallis test with pairwise two-sided Wilcoxon test and a Benjamini & Hochberg correction were used for 5f. ***$p < 0.001$; **$p < 0.01$; *$p < 0.05$; ns: $p > 0.05$. Source data are provided as a Source Data file. Schemes were taken from Franklin and Paxinos Brain Atlas, 3rd edition.

overactivation of this pathway in LID. Even though the exact link between cerebellar stimulation, FosB/ΔFosB, and corticostriatal plasticity remains to be determined, these results indicate a widespread normalization in the cerebello-striato-cortical motor system and suggest that the cerebellar stimulation acts on core mechanisms of LID.

## Discussion

We used optogenetic stimulations, extracellular recordings, and chemogenetic inhibition to investigate the role of PC in CrusII, the orolingual region of the cerebellum, in the alleviation of orolingual levodopa-induced dyskinesia (LID). Previous clinical

studies found a reduction of LID severity using cerebellar rTMS in PD patients[17,19,20,64]. However, the precise mechanisms, pathways and cell-types responsible for this beneficial effect remained unknown. In the present study, we first show that CrusII PC opto-stimulation corrects, or even prevents, severe orolingual dyskinesia exhibited by chronically levodopa-treated PD mice. These results demonstrate a direct involvement of PC in the anti-dyskinetic effect of the cerebellum. This beneficial effect led to complete alleviation of orolingual dyskinesia and thus was stronger than observed in patients where rTMS was applied bilaterally over the hemispheres of the cerebellum[18,19]. However, the effect of rTMS on cerebellum is not yet well understood: rTMS may only indirectly activate PC and its efficacy may be constrained by the difficulty to target the optimal depths of the cerebellar cortex[86]. In this study, we found that the beneficial effect of cerebellar stimulation in CrusII, which hosts dense projections from the orolingual area[57], is mainly observed on the orolingual LID, suggesting a correspondence between the cerebellar somatotopy and functional impact of cerebellar stimulation. Similarly, different subtypes of LID are associated with different patterns of striatal FosB/ΔFosB expression levels, consistent with striatal somatotopy[38]. The efficacy of cerebellar rTMS in patients should thus strongly depend on the site of stimulation. Cerebellar rTMS has been reported to induce changes within the cerebellar cortex[87]. However, our work demonstrates a normalization of the neuronal activity in a wide motor network following cerebellar stimulation, reveals a contribution of the cerebello-thalamo-striatal pathway in mediating the effect of PC

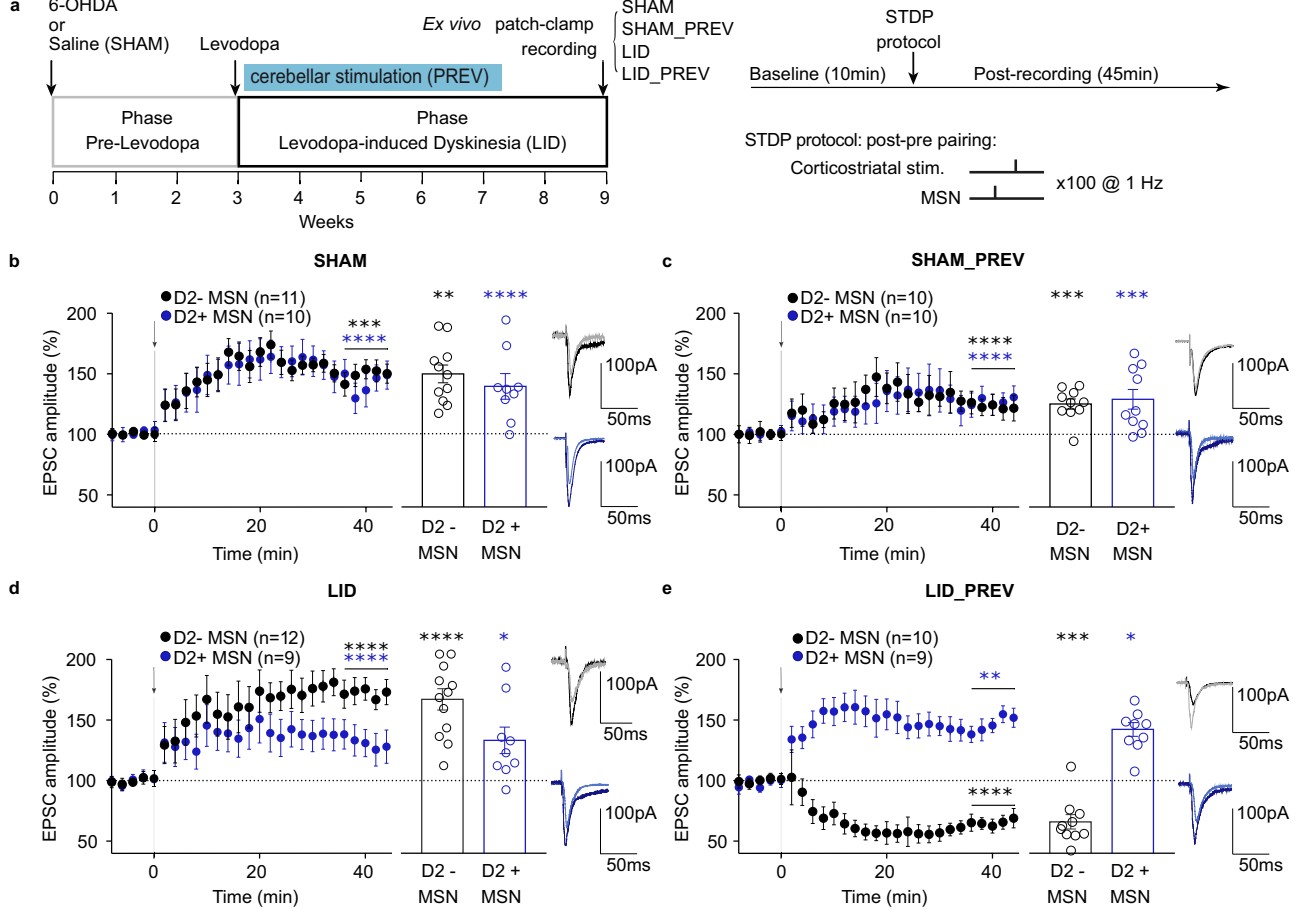

**Fig. 6 Spike-timing dependent plasticity produces LTD instead of LTP in D1-expressing neurons following Purkinje cell stimulation. a** *Left:* Experimental timeline. SHAM mice: 6 weeks of levodopa treatment. Preventive SHAM mice (SHAM_PREV): 6 weeks of levodopa treatment + 4 weeks of cerebellar stimulation. Dyskinetic mice (LID): 6 weeks of levodopa treatment. Preventive mice (LID_PREV): 6 weeks of levodopa treatment + 4 weeks of cerebellar stimulation. Ex vivo experiments were realized on mice subjected to 4 conditions, i.e. SHAM, SHAM_PREV, LID and LID_PREV. *Right:* STDP pairings: A single spike evoked in the recorded MSN (post) was paired with a single cortical stimulation (pre); pairings were repeated 100 times at 1 Hz. **b–e** Averaged time courses of corticostriatal STDP in D1-MSNs and D2-MSNs induced by 100 post pre pairings. **b** In SHAM, LTP induced by 100 post–pre pairings in D1-MSNs and D2-MSNs ($n = 21$). **c** In SHAM_PREV, LTP induced by 100 post–pre pairings in D1-MSNs and D2-MSNs ($n = 20$). **d** In LID, LTP induced by 100 post–pre pairings in D1-MSNs ($n = 12$) and D2-MSNs ($n = 9$). **e** In LID_PREV, LTP induced by 100 post–pre pairings in D1-MSNs ($n = 10$) and the same protocol induced LTD in D2-MSNs ($n = 9$). Synaptic strength was determined 34–44 min after pairings. Bar graphs represent the average of all STDP experiments, and each point represents the percentage of change in excitatory postsynaptic current (EPSC) amplitude at 34–44 min after STDP pairings in a single STDP experiment. Insets correspond to the average EPSC amplitude at baseline (grey and light blue) and at 34–44 min after STDP pairings (black and dark blue). Error bars represent the SEM. **p < 0.01; ***p < 0.001; ****p < 0.0001 by one sample two-sided *t* test. Source data are provided as a Source Data file.

stimulation, and shows that it normalizes the expression of striatal FosB/ΔFosB, causally linked to LID. Overall, our results indicate that PC stimulation exerts long-range effects and acts on core mechanisms of LID outside of the cerebellum.

Our study further characterizes the alterations occurring in the cerebello-thalamo-cortical and cerebello-thalamo-striatal pathways during dyskinesia. The increased activity of the primary motor cortex in LID observed in our study is consistent with previous findings in rodents and humans[21,24]. The gradual ramping of activity observed in M1, compared to the strong and steady expression of severe LID, might reflect slower mechanisms such as changes in plasticity and/or connectivity which might not be primarily involved in the generation of the abnormal movements but rather gradually lock the motor circuits in the dyskinetic state. Restoring M1 activity might therefore represents a key component to unlock motor circuits from their pathological state. M1 influences the CN and the cerebellar cortex through pontine nuclei[48,88,89]. This observation fits the decreased activity observed

in the IN and DN in LID. In addition, cerebellar nuclei neurons exhibited increased irregularity discharge, and such cerebellar anomalies have been implicated in other motor disorders, such as tremor[73], ataxia[90] and dystonia[71,72]. The irregular activity found in cerebellar nuclei neurons could thus contribute to LID. A similar decrease of activity is observed in PF neurons. This change could result from a decreased cerebellar entrainment of PF through the cerebello-parafascicular connections[51–54]. However, our results failed to demonstrate consistent modulations in the cerebellar region of the motor thalamus in LID or following cerebellar stimulation, suggesting that this structure might be less recruited for the expression and alleviation of LID following cerebellar stimulation. In patients, some authors have indeed suggested that cerebellar stimulation using rTMS might act primarily on the sensory afferent volley reaching M1 rather than direct effects on the motor afferents through VAL[27]. Overall, the changes in activity observed in LID are likely inter-dependent since they were all prevented by cerebellar stimulation.

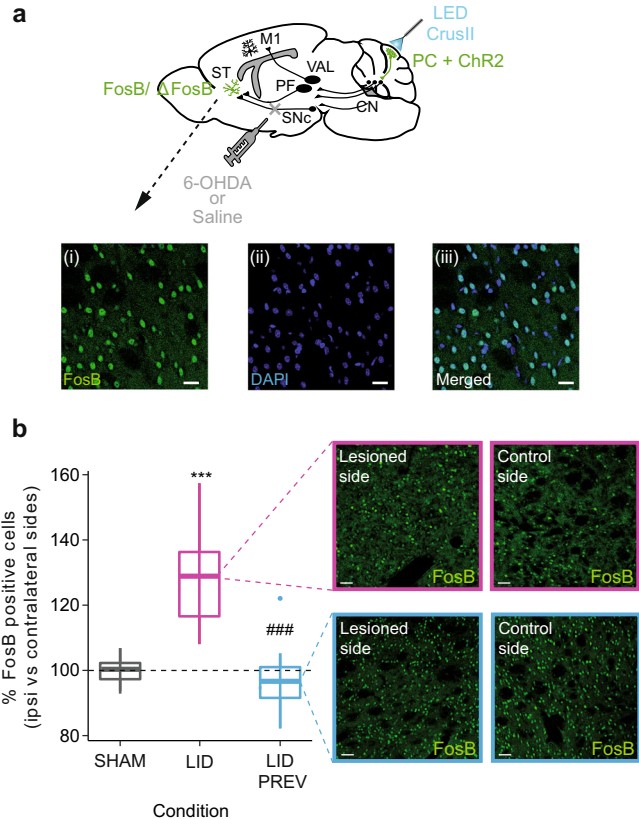

**Fig. 7 Striatal overexpression of the dyskinetic marker FosB/ΔFosB is restored by Purkinje cell stimulation. a** Sagittal schematic showing neurons in the striatum expressing the dyskinetic marker FosB/ΔFosB (green). The cerebello-thalamo-cortical and cerebello-thalamo-striatal pathways are represented in mice expressing ChR2-YFP in Purkinje cells (PC + ChR2, green) as well as the injection site 6-OHDA or saline (gray). ST: Striatum; SNc: substantia nigra *pars compacta*; M1: Primary motor cortex, VAL: Ventroanterior-ventrolateral complex of the thalamus, PF: Parafascicular nucleus of the thalamus, CN: cerebellar nuclei, CrusII: Crus2 of the ansiform lobule. Insets: Postmortem histology showing FosB-expressing neurons (i), DAPI (ii), and merged (iii). Scale bars: 20 μm. **b** Boxplots showing the ratio of cells expressing FosB between the striatum ipsilateral to the lesion and the striatum contralateral to the lesion in percentage (%) in the 4 different conditions (LID, magenta, $N = 10$; SHAM, gray, $N = 10$; and LID_PREV, blue, $N = 10$). Horizontal bars in boxplots represent the median. Magenta inset: Postmortem histology showing FosB-expressing neurons in dyskinetic animals in the striatum ipsilateral to the lesion (*left box*) and in the striatum contralateral to the lesion (*right box*). Scale bars: 20 μm. Blue inset: Postmortem histology showing FosB-expressing neurons in preventive animals in the striatum ipsilateral to the lesion (*left box*) and in the striatum contralateral to the lesion (*right box*). Scale bars: 20 μm. Boxplots represents the lower and the upper quartiles as well as the median. Vertical lines represent the median ± std. Isolated points represent outliers of the distribution. Two-sided student t test. ***$p < 0.001$; **$p < 0.01$; *$p < 0.05$. *Compared to SHAM; # compared to LID. Source data are provided as a Source Data file. See also Table S14.

Cerebellar stimulation produces alternate periods of silence and increased "rebound" activity in CN[91]. Previous evidence demonstrated that this rebound activation is propagated in the forebrain motor network[78]. This phenomenon might explain the normalization of activity observed in CN and PF following cerebellar stimulation. The chemogenetic inhibition of the cerebello-parafascicular neurons reduced the beneficial impact of cerebellar stimulation suggesting an important contribution of the cerebello-thalamo-striatal pathway. Examination of the collaterals of these neurons revealed only sparse collaterals to the motor thalamus suggesting a primary role of the thalamostriatal over thalamocortical projections, further reinforced by the lack of modulation observed in VAL following cerebellar stimulation. The striatum is indeed playing a core role in LID generation notably through the overactivity of the direct pathway within the striatum[11,13,15]. This overactivity could result from an excessive potentiation at the corticostriatal synapses of direct pathway neurons[14,43]. Indeed, abnormal LTP at corticostriatal synapses of D1-expressing MSNs has been observed in many different studies in LID and is thought to be one of the major causes locking the basal ganglia system in a pathological state[14,42,43]. As cerebellar stimulation exerts long-term beneficial impacts by preventing the appearance of LID and so, up to 2 weeks after stimulation stopped, cerebellar stimulation might alter striatal plasticity, as previously demonstrated[49], leading to these long-term beneficial effects. We found that preventive cerebellar stimulation converted the corticostriatal LTP into LTD in direct pathway MSNs in preventive mice. This suggests that this stimulation promoted corticostriatal LTD over LTP in the direct pathway and may therefore circumvent the excessive potentiation occurring in LID[43]. We also found upregulation of striatal FosB/ΔFosB in LID, consistent with previous studies[35,36,38,40]. Since FosB/ΔFosB overexpression suffices to trigger LID, the normalization of striatal FosB/ΔFosB levels by cerebellar stimulation may explain the suppression of LID. FosB/ΔFosB has been shown to be mainly expressed in D1-expressing striatal neurons[30,36,38], suggesting that cerebellar stimulation can modulate transcriptional activity in D1-MSNs, probably through projections of IN to PF and the striatum[54]. These changes of transcriptional activities might also be related to the change in corticostriatal plasticity observed in our study. Through its cerebello-thalamo-striatal pathway, cerebellar stimulation might alter striatal plasticity, as observed in other studies[49] and by promoting LTD at corticostriatal plasticity in D1-MSNs, cerebellar stimulation might downregulate the expression of FosB/ΔFosB through the downregulation of other transcription factors involved in D1 signaling pathway and LID, such as ERK1/2 (Fig. 8). Overall, these results show that our protocol of cerebellar stimulation induces profound changes in the striatal function leading to long-lasting benefits. The persistence of the beneficial effects of this protocol after its end indicates that it recruits a long-term plasticity that could be harnessed for the improvement of LID in PD patients.

Consistent with our finding that cerebellar stimulation may exert a transient therapeutic effect, stimulation of the output pathways of the cerebellum has been recently shown to reduce tremor and ataxia[73,92]. Therefore, improving the experimental approaches aimed at stimulating cerebellar Purkinje cells or cerebellar nuclei neurons may benefit to multiple motor disorders. Finally, our work confirms the necessity to study LID as a network disorder involving abnormal signaling between the basal ganglia, cerebral cortex, thalamus and cerebellum[16,44].

## Methods

**Animals and protocol**. L7-ChR2;WT mice[55] were used for in vivo experiments, L7-ChR2;Drd2-GFP mice were used for ex vivo experiments. Animals were housed 1–3 per cage on a standard 12-hour light/dark cycle with *ad libitum* access to water and food and with a constant humidity of 40% and temperature of 22 °C. All behavioral manipulations took place during the light phase. All experiments were performed on mice aged 6–9 weeks, of either sex (35-45 g), from the Institut de Biologie de l'Ecole Normale Supérieure, Paris, France and in accordance with the recommendations contained in the European Community Council Directives. This project (referenced APAFIS#1334-2015070818367911 v3 and APAFIS#29793-202102121752192 v3) has been approved by the ethical review board of the Ministry of Higher Education, Research, and Innovation by the Cell responsible for the Use of Animals for Scientific Purposes (AFiS). All animals followed a 9 to 10-weeks experimental protocol. After surgical intervention, mice were carefully

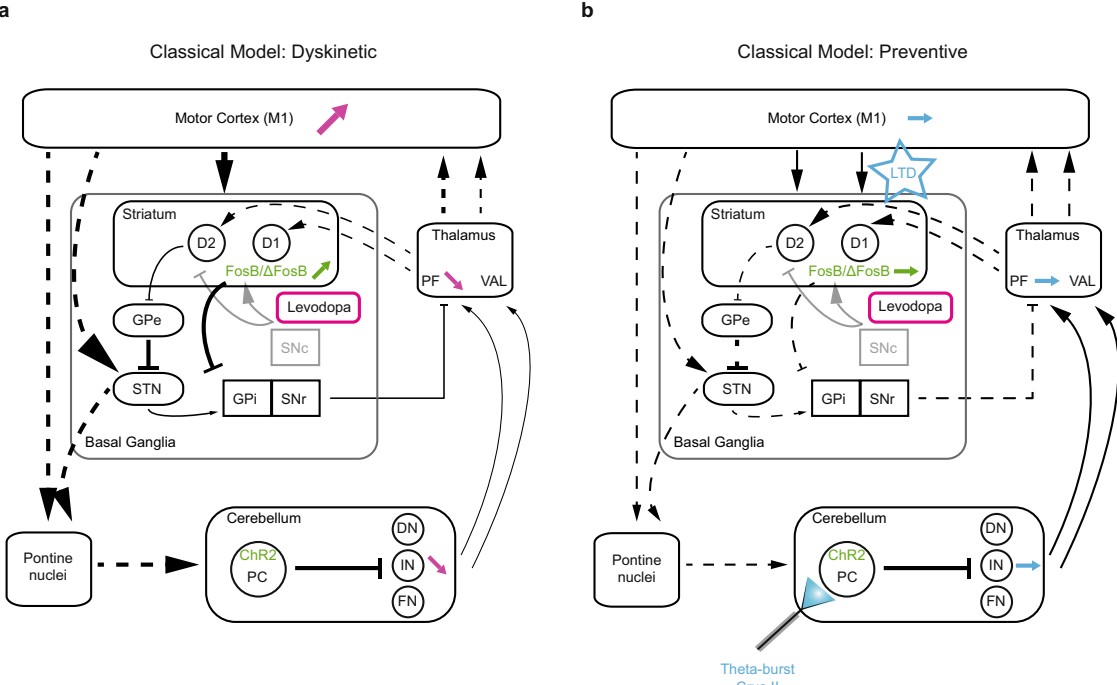

**a** Classical Model: Dyskinetic

**b** Classical Model: Preventive

**Fig. 8 Preventive PC stimulations normalize aberrant activity in the cerebello-thalamo-striatal pathway associated to dyskinesia.** Schematic representation of the cerebello-thalamo-cortical, cerebello-thalamo-striatal, and basal ganglia-thalamo-cortical pathways and their modulations observed in dyskinetic (**a**) and preventive mice (**b**). Schematic representation of the cerebello-thalamo-cortical, cerebello-thalamo-striatal, and basal ganglia-thalamo-cortical pathways and their modulations in (**a**) levodopa-induced dyskinesia (magenta arrows) and (**b**) in preventive mice (blue arrows). Green arrows represent modulations of expression in FosB/ΔFosB in the dorsolateral striatum, black arrows represent excitatory projections, and blunted arrows represent inhibitory connections. Dotted lines represent unknown interactions, thick lines represent increased output, and thin lines represent decreased output. Faded projections represent the loss of the pathway. D1: D1-expressing striatal projection neurons; D2: D2-expressing striatal projection neurons; GPe globus pallidus pars externus; GPi globus pallidus pars internus; SNc substantia nigra pars compacta; SNr substantia nigra pars reticulata; STN subthalamic nucleus; VAL ventroanterior-ventrolateral complex of the thalamus; PF parafascicular nucleus of the thalamus; M1: primary motor cortex; PC Purkinje cells; DN dentate cerebellar nucleus; IN interposed cerebellar nucleus; FN fastigial cerebellar nucleus; LTD long-term depression. Adapted from[5].

monitored during 1-1.5 weeks following a nursing protocol adapted from[59] to reduce post-surgery lethality. All conditions included, the survival rate reached 80%. After 3 weeks, animals received daily intraperitoneal (I.P) injections of L3,4-dihydroxyphenylalanine methyl (L-DOPA, 3 days at 3 mg/kg, then 6 mg/kg, Sigma-Aldrich) and the peripheral DOPA decarboxylase inhibitor bensezaride hydrochloride (12 mg/kg, Sigma-Aldrich) for 6 weeks (0.1 mL/10 g body weight). Animals were separated into 4 groups: LID mice, received daily L-DOPA injection alone, LID_PREV, received daily theta-rhythm cerebellar "preventive" stimulation from the first day of levodopa administration (3 mg/kg), LID_CORR, received daily theta-rhythm cerebellar "corrective" stimulation after 2 weeks of L-DOPA injection (6 mg/kg). Cerebellar stimulation was applied 30 min after L-DOPA injection to maximize the differential effects between cerebellar stimulation and L-DOPA. Finally, SHAM mice, received daily L-DOPA and either no stimulation or preventive or corrective theta-rhythm cerebellar stimulation. Cerebellar stimulation was stopped simultaneously in all conditions and the last two weeks were assessed for long-term anti-dyskinetic effects. Behavioral assessment was performed every week (every 7days exactly), starting on the first day of L-DOPA 6 mg/kg (see below). Electrophysiological measurements were performed every week as well after behavioral assessments.

**Surgical procedures**. All surgical procedures were performed at 6–9 weeks of age. After subcutaneous (S.C) administration of buprenorphine (0.06 mg/kg), animals were anesthetized with isoflurane (3%) and maintained with 0.5%-1.0% inhaled isoflurane. Mice were placed in a stereotaxic frame (David Kopf Instruments, USA) and pretreated with desipramine (25 mg/kg, I.P) (Sigma-Aldrich). Small holes were drilled over the left medial forebrain bundle (MFB: −1.2 AP, 1.3 ML, −3.75 mm DV) and either over the left oral motor cortex (M1: + 2.2 AP, −2.2 ML, −1.5 mm DV), the left parafascicular nucleus (PF: −2.3 AP, −0.75 ML, −3.5 mm DV), and the left ventroanterior-ventrolateral complex of the thalamus (VAL: −1.40 AP, −1.40 ML, −3.50 mm DV) or over the right fastigial nucleus (FN: −6.4 AP, + 0.85 ML, −3.25 mm DV), interposed nucleus (IN: −6.4 AP, + 1.6 ML, −3.25 mm DV), and dentate nucleus (DN: −6.2 AP, + 2.3 ML, −3.3 mm DV). The left MFB was injected with either 1 µL of 6-Hydroxydopamine hydrochloride (6-OHDA, (Sigma-Aldrich) 3.2 µg/µL free-base in a 0.02% ascorbic acid solution (Sigma-

Aldrich)) (for the parkinsonian model) or 1 µL vehicle (ascorbic acid for SHAM animals) at a rate of 0.1 µL/min, after which the syringe was left in place for 10 min. An additional hole was drilled over the left cerebellum for placement of a skull screw (INOX A2, Bossard) coupled to the ground wire. A final craniectomy centered over the left Crus II (−6.3 to −7.3 mm AP, + 3.0 to +4.2 mm ML) was performed, without removing the dura to prevent damage to the cerebellar cortex. A 2.88 mm$^2$ LED (SMD chip LED lamp, Kingbright, USA) was then cement to the skull over CrusII. Recording electrodes were slowly lowered through the craniectomy at the wanted coordinates. The ground wire was clipped to the recording board using pins (Small EIB pins, Neuralynx, Dublin, Ireland) and the entire recording device was secured with dental cement (Metabond) and dental acrylic (Pi-Ku-Plast HP 36, monomer and polymer, Bredent, Germany). All animals were given anti-inflammatory Metacam S.C (Metacam 2 mg/mL, Boehringer Ingelheim) for postoperative analgesia and sterile glucose-saline solutions S.C[59] (Glucose 5%, Osalia). Parkinsonian-like animals were closely monitored for 1-1.5 weeks following surgery, if needed mice's cages were kept on a heating pad, animals received several glucose-saline injections daily and were fed a mixt of bledine (Blédina, Danone, France) and concentrated milk.

**Behavior**

*Open field*. Animals were monitored in a 38 cm diameter open field (Noldus, The Netherlands) once a week for 5 min during 9 weeks. The mice were monitored with a camera (Allied Vision Prosilica GigE GC650, Stemmer Imaging) placed directly above to assess periods of inactivity/activity. The video acquisition was made at 25 Hz frequency and at a resolution of 640 by 480 pixels. DeepLabCut method was used for the analysis of locomotor activity. Seven points of interest were labeled: nose, left ear, right ear, basis of tail, end of tail, red led of the headstage, and green led of the headstage. Labels were manually applied to the desired points on 500 frames. The position of the mouse head has been reconstructed from the barycenter of the points: nose, left ear, right ear, red led of the headstage, and green led of the headstage weighted by their likelihood. Then, to this head's trajectory, a cubic smoothing spline fit was applied. For the analysis of locomotion, the periods of activity were isolated from the periods of inactivity by thresholding the speed of the head point at 1 cm/s.

*AIMs assessment.* After a 3-weeks pre-L-DOPA phase, daily IP injections of levodopa begun. In rodent models of LID, abnormal involuntary movements (AIMs) are considered the behavioral and mechanistic equivalent of LID in PD patients[14,56]. To assess AIMs in our mouse model, a modified scale was used[56]. Each mouse was placed in a glass cylinder surrounded by 2 mirrors to detect AIMs in every angle and was recorded with a video camera (Allied Vision Prosilica GigE GC650, Stemmer Imaging) for 4 min every 20 min over a 2-h period, starting 20 min before L-DOPA injection. *Post-hoc* scoring was performed for 2 min every 20 min. The mice were then evaluated a total of 8 time points during the whole recording. Movements were identified as dyskinetic only when they were repetitive, affected the side of the body contralateral to the lesion, and could be clearly distinguished from naturally occurring behaviors such as grooming, sniffing, rearing, and gnawing. Specifically, AIMs were classified into three categories based on their topographic distribution: axial, forelimb, and orolingual. Forelimb dyskinesia are defined as hyperkinetic and/or dystonic movements of the contralateral forelimb on the sagittal or frontal plane. Axial AIMs are considered a twisted posture of the neck and upper body towards the side contralateral to the lesion. Finally, orolingual AIMs are defined as repetitive and empty chewing movements of the jaw, with or without tongue protrusion, which do not correspond to normal movements of the mouth. Orolingual LID has been used as a readout of the beneficial impacts of PC stimulation based on the topographical organization of both the cerebellum and LID. Each subtype of AIMs were scored on a severity scale from 0 to 4 where: 0 = absent; 1 = occasional occurrence, less than half of the observation period; 2 = frequent occurrence, more than half of the observation time; 3 = continuous but interrupted by sensory stimuli; and 4 = continuous and not suppressible by sensory distraction[56]. AIMs scores were summed per time point (8 time points total per mouse) and averaged per week per condition. AIMs were scored starting on the first day of L-DOPA 6 mg/kg (thus day 4 of the protocol) in all conditions and then scored every week (Fig. 1a). Non-parametric Kruskal-Wallis test with pairwise two-sided Wilcox test and a Benjamini & Hochberg correction were used to compared the averaged scores across time and between conditions for peak-dose and average off-period dyskinesia scores.

**Optogenetic stimulations.** To stimulate PC on CrusII cerebellar cortex, 3 micro LEDs (1.6 × 0.6 mm, SMD chip LED lamp, Kingbright, USA) emitting blue light with a dominant 458 nm wavelength, were soldered together to cover the full extent of the stimulated area. A small piece of coverslip glass was glued to the bottom of the LEDs to prevent heat brain damage. Two insulated power wires were connecting the LEDs to allow connection with a stimulating cable (New England Wire Technologies, Lisbon) coupled to a LED driver (Universal LED Controler, Mightex). Stimulated mice (LID_PREV, LID_CORR) and appropriate SHAM (SHAM_PREV and SHAM_CORR) received daily train of 20 ms stimulations delivered at 8.33 Hz (theta-rhythm) and at 16 mW/mm² irradiance for 2x40 seconds separated by a 2 min period, every day, 30 min after L-DOPA injection as used in[19,93,94].

**In vivo freely moving chronic recordings.** To record cell activity, we used bundles of electrodes consisting of nichrome wire (0.005 inches diameter, Kanthal RO-800) folded and twisted into bundles of 4–8 electrodes. Prior to surgical intervention, the bundles were pinned to an electrode interface board (EIB-16; Neuralynx, Dublin, Ireland) according to the appropriate coordinates of the targeted structures. The microwires of each bundle were connected to the EIB-16 with gold pins (Neuralynx, Dublin, Ireland). The entire recording device was secured by dental cement. The impedance of every electrode was set to 250–350 kΩ using gold-plating (Cyanure-free gold solution, Sifco, France). Electrical signals were acquired using a headstage and amplifier from TDT (RZ2, PZ5, Tucker-Davis Technologies, Alachya, FL, USA), filtered, amplified, and recorded on Synapse System (Tucker-Davis Technologies, Alachya, FL, USA). Spike waveform were filtered at 2 Hz high-pass and 200 Hz low-pass and digitized at 25 kHz. The experimenter manually set a threshold for storage and visualization of electrical events.

During recording sessions, after a 5-min open field and a 20-min recording baseline, L-DOPA (6 mg/kg) was injected IP and spontaneous activity was recorded for 1h30 and stimulated using a LED driver (Universal LED Controler, Mightex), an automatized commutator (ACO32 SYS3-32-CH motorized commutator, Tucker-Davis Technologies, Alachya, FL, USA), and controlled by TTL pulses from our behavioral monitoring system (RV2, Tucker-Davis Technologies, Alachya, FL, USA). At the end of recording sessions, animals were detached and returned to their home cages.

Single units were identified offline by manual spike sorting performed on Matlab R2006a (Mathworks, Natick, MA, USA) scripts based on k-means clustering on principle component analysis (PCA) of the spike waveforms[95]. Recorded units were spike sorted over the entire recording time (90 min) assuring that the sorted units are present over the entire recording session. One cluster was considered to represent a single-unit if the unit's spike waveform was different from other units on the same wire, in 3D PCA space. The unit's firing activity was analyzed from all structures using R Studio software 4.1.2. No statement is made whether the same cells were recorded across the 9-weeks session. Because many different events (levodopa administration, peak-dose effect, cerebellar stimulation) might alter the firing rate of the recorded units across one single session, averaging the firing rate over the entire session was excluded. Therefore, only a 10-minutes

recording window corresponding to the peak of L-DOPA plasmatic concentration and thus, on LID severity and right after cerebellar stimulation has been selected. For display purposes, the firing rate of single units was averaged per condition and per group of weeks using boxplots. Boxplots show the median rate in Hz, represented by horizontal bars, over 4 categories of weeks: first boxplot represents the 2nd and 3rd week of the protocol (pre-L-DOPA), second boxplot represents the 4th and 5th weeks when levodopa treatment has started as well as cerebellar stimulation for preventive mice, third boxplot represents the 6th and 7th weeks when corrective mice start receiving cerebellar stimulation, the last boxplot represents the 8th and 9th weeks of the protocol when stimulation stops and long-term effects of cerebellar stimulation are visible. Welch Anova with two-sided Games Howell post-hoc test or one-way Anova's with two-sided Tukey post-hoc test were performed based on the results of Levene test to compared the averaged firing rate between conditions in CN whereas linear model ANOVA with the mice randomly distributed and two-sided Tukey's Post-Hoc test were performed to compared the averaged firing rate between conditions in M1, PF, and VAL.

**Neuroanatomical tracing.** Mice were injected with 100nL of retrograde AAV-Syn-GFP (Titer: $1 \times 10^{13}$ GC/mL, Lot#V16660, Addgene) in PF ($N = 6$). For synaptophysin labeling and axon collaterals, 70nL of AAV8.2-hEF1a-DIO-synaptophysin-GFP (Titer: $2.19 \times 10^{13}$ GC/mL, Massachusetts General Hospital) together with 50 nL of AAV1.CAG.Flex.tdTomato.WPRE.bGH (Titer: $7.8 \times 10^{13}$ GC/mL, Lot #CS0923, Upenn Vector) were injected in IN, DN, and FN, in combination with 150 nL of retrograde AAV-Cre-EBFP (Titer: $1 \times 10^{13}$ GC/mL, Lot #V15413, Addgene) injection in PF. Mice were perfused (see below) 15-25 days after injections to allow the expression of the AAVs. Brains were sliced entirely at 90 μm using a vibratome (Leica VT 1000 S), and mounted on gelatin-coated slides, dried and then coverslipped with Mowiol (Sigma). Slices were analyzed and imaged using a confocal microscope (SP8, Leica), and images were edited and analyzed using FIJI/ImageJ-win 64 v1.53q. One mouse was excluded from this experiment for off-target injection. Data are represented as mean ± SEM.

**Chemogenetic experiment.** Mice were injected with inhibitory DREADDs pAAV8-hSyn-DIO-hM4DimCherry (Titer: $2.9 \times 10^{13}$ GC/mL, Vol: 100 μL, Lot: v54499; Addgene) in IN, DN, and FN (150nL per nucleus) contralaterally-to-the-lesion, in complementation with 300nL CAV2-Cre-GFP (Titer: $6.4 \times 10^{12}$, dilution 1/10, Plateforme de Vectorologie de Montpellier) viral infusion in ipsilateral-to-the-lesion PF. CAV2-Cre-GFP injections' surgeries were performed 1 week before the injections of pAAV-hSyn-DIO-hM4DimCherry and the MFB lesion. After 3 weeks, to allow good expression of the viruses and to reproduce our Parkinsonian model, the animals started the levodopa treatment for 6 weeks and the cerebellar stimulation for 4 weeks (LID_PREV). The severity of their orolingual LID was scored 40 min after levodopa injection (at the peak dose) and right after cerebellar stimulation. Scores were averaged across the 4 weeks of preventive stimulation. For neuronal modulation of animals expressing DREADDs, Clozapine N-oxide (1.25 mg/kg, Tocris Bioscience) was diluted in saline and injected I.P 10 min prior L-DOPA. Control group was injected with saline. Slices were analyzed and imaged using a confocal microscope (SP8, Leica). Images were edited and cells were counted using FIJI/ImageJ. mCherry positive cells in the CN were counted from three sections per animals in both controls, a.k.a. non-virally injected mice, and CNO injected mice. Experimenters were blinded to the experimental groups. Anova with two-sided Tukey HSD post-hoc test was used for Fig. 5e and non-parametric Kruskal-Wallis test with pairwise two-sided Wilcoxon test and a Benjamini & Hochberg correction were used for Fig. 5f. ***$p < 0.001$; **$p < 0.01$; *$p < 0.05$; ns: $p > 0.05$.

**Perfusion, immunohistochemistry, microscopy, and cell counting.** Mice were anesthetized with ketamine/xylazine I.P and transcardially perfused with 4% paraformaldehyde in PBS (Formalin solution, neutral buffer 10%, Sigma-Aldrich). Brains were dissected and post-fixed for 24 h in 4% PFA, 24 h in 20% sucrose (Merck) and 24 h in 30% sucrose. Coronal 20 μm sections were cut using a freezing microtome (Leica) and mounted on Superfrost glass slides (Superfrost Plus, Thermo Fisher) for imaging. The good expression of ChR2(H134R) in PC has been verified by (i) genotyping the animals prior to experiment; (ii) detecting the emission light emitted by the Yellow Fluorescent Protein (YFP) attached to ChR2(H134R) using a confocal.

For immunohistochemistry, the tissue was blocked with 3% normal donkey serum (NDS, JacksonImmunoResearch) or normal goat serum (NGS, JacksonImmunoResearch) and permeabilized with 0.1% Triton X-100 (Sigma-Aldrich) for 2 h at room temperature on a shaker. Primary antibodies: Guinea pig anti-TH (Synaptic System, 1:500) and Rabbit anti-FosB (Santa Cruz, 1:100) were added to 1% NDS or NGS and incubated overnight at 4 °C on a shaker. Secondary antibodies: donkey anti-Guinea Pig Cy3 (JacksonImmunoResearch, 1:400) and goat anti-Rabbit Alexa 488 (JacksonImmunoResearch, 1: 200) were added in 1% NDS/NGS for 2 h at room temperature on a shaker. The slices were then washed, incubated with Hoechst (Invitrogen, ThermoFisher scientific, 1:10 000), and mounted onto slides for visualization and imaging. The color reaction was acquired under a dissecting microscope (Leica) and 5, 20, 40 or 64x images were taken. For FosB/ΔFosB quantification, images were taken using confocal microscope (SP8,

Leica) with 10 and 40x objective. Exposure time were matched between images of the same type. Individual images were stitched together to produce an entire coronal image of both striatum. One section of the dorsolateral part of the middle striatum (~ AP: 0.50 mm; DV: −3.0 mm; ML: + 2.5 mm) per animal was used for quantification and FIJI/ImageJ software was used to count cells using manual counting. Several experimenters performed FosB/ΔFosB cell counts on a subset of animals and all experimenters were blinded to the experimental groups. Two-sided student $t$ test was used for statistics. The extent of the dopaminergic lesion was quantified using an optical density analysis on TH staining between the two striatum. One section of the middle striatum per animal (bregma 0.50 – bregma 0.25) was used for TH quantification. Mean density of fluorescence of each striatum was normalized on the mean density of fluorescence of the ipsilateral corpus collasum. All animals were dopaminergically depleted >50% and thus included in the study. Some animals were excluded from the study for (i) lack of LID expression in the levodopa-treated condition (LID; $N = 2/14$) and (ii) for misplacing of the electrodes (all conditions; $N = 18/53$). At the end of the protocol, the animals were subsetted in the different experiments: FosB/ΔFosB cell counts, TH analysis, and ex vivo experiment.

**Ex vivo whole-cell patch-clamp electrophysiology.** SHAM, SHAM_PREV, LID, and LID_PREV mice were anaesthetized using isofluorane and their brain removed from the skull. Horizontal striatal slices (270 μm-thick), containing the dorsal lateral striatum, were cut using a VT1000S vibratome (VT1000S, Leica Microsystems, Nussloch, Germany) in ice-cold oxygenated solution (ACSF: 125 mM NaCl, 2.5 mM KCl, 25 mM glucose, 25 mM NaHCO3, 1.25 mM NaH2PO4, 2 mM CaCl2, 1 mM MgCl2, 1 mM pyruvic acid). Slices were then incubated at 32–34 °C for 60 min before returning to room temperature in holding ACSF. For whole-cell recordings, borosilicate glass pipettes of 4–8 MΩ resistance were filled with a potassium gluconate-based internal solution consisting of (in mM): 122 K-gluconate, 13 KCl, 10 HEPES, 10 phosphocreatine, 4 Mg-ATP, 0.3 Na-GTP, 0.3 EGTA (adjusted to pH 7.35 with KOH, osmolarity 296 ± 3.8 mOsm). Signals were amplified using with EPC10–2 amplifiers (HEKA Elektronik, Lambrecht, Germany). All recordings were performed at 32–34 °C, using a temperature control system (Bath-controller V, Luigs&Neumann, Ratingen, Germany) and slices were continuously superfused with extracellular solution at a rate of 2 ml/min. Recordings were sampled at 10 kHz, using the Patchmaster v2 × 32 program (HEKA Elektronik). D2⁺-MSNs were visualized under direct interference contrast with an upright BX51WI microscope (Olympus, Japan), with a 40x water immersion objective combined with an infra-red filter, a monochrome CCD camera (Roper Scientific, The Netherlands) and a compatible system for analysis of images as well as contrast enhancement.

Spike-timing-dependent plasticity (STDP) protocols of stimulations were performed with one concentric bipolar electrode (Phymep, Paris, France; FHC, Bowdoin, ME) placed in the layer 5 of the somatosensory cortex while whole-cell recording MSN in the dorsolateral striatum. Electrical stimulations were monophasic, at constant current (ISO-Flex stimulators, AMPI, Jerusalem, Israel). Currents were adjusted to evoke 100–300 pA EPSCs. STDP protocols consisted of pairings of post- and presynaptic stimulations separated by a specific time interval (~20 ms); pairings being repeated at 1 Hz. The postsynaptic stimulation of an action potential evoked by a depolarizing current step (30 ms duration) in the recorded MSN preceded the presynaptic cortical stimulation, in a post-pre pairing paradigm. Post–pre pairings was repeated 100 times at 1 Hz (Fig. 6a1). Recordings on neurons were made over a period of 10 min at baseline, and for at least 40 min after the STDP protocols; long-term changes in synaptic efficacy were measured in the last 10 min. Experiments were excluded if the mean input resistance (Ri) varied by more than 20% through the experiment. Off-line analysis was performed with Fitmaster (HEKA Elektronik), IGOR Pro 6.0.3 (WaveMetrics, Lake Oswego, OR, USA). Statistical analyses were performed with Prism 7.00 software (San Diego, CA, USA). All results are expressed as mean ± SEM. Statistical significance was assessed in one-sample t tests, unpaired t tests as appropriate, using the indicated significance threshold (p).

**Statistics and reproducibility.** Unless otherwise stated, data are presented as boxplots (from boxplot function in R Studio software 4.1.2). Boxplots represent the median (horizontal bars) as well as the lower and the upper quartiles. Vertical lines represent the median ± std. Isolated points represent outliers of the distribution and are included in statistical analysis. Statistical differences for all behavioral experiments were assessed using Kruskal-Wallis test with pairwise two-sided Wilcoxon test and Benjamini & Hochberg correction. ***$p < 0.001$; **$p < 0.01$; *$p < 0.05$; ns: $p > 0.5$. * represent comparison to SHAM, # represents comparison to LID. Statistical differences for all in vivo electrophysiological experiments were assessed using either Welch Anova with two-sided Games Howell post-hoc test or one-way Anova's with two-sided Tukey post-hoc test based on the results of the Levene test. Statistical significance in ex vivo experiments was assessed in one-sample two-sided $t$ tests, unpaired $t$ tests as appropriate, using the indicated significance threshold (p). ***$p < 0.001$; **$p < 0.01$; *$p < 0.05$; ns: $p > 0.5$. All statistical analysis were performed using R and R Studio software 4.1.2. No statistical method was used to predetermine sample size. All experiments were performed several times (3 to 5x) and by several experimenters and all replicates led to similar results. Anatomical tracing and chemogenetic experiments were performed only once but on several independent animals. Only the data collection per se for the in vivo electrophysiological and chemogenetic experiments were not performed by

experimenters blinded to the condition as the different conditions required different treatments, however, the data analysis of these experiments was performed by experimenters blinded to the experimental groups. When the experiments were performed by several experimenters, statistical analysis were performed with randomization of the experimenters. In all experiments, experimenters were blinded to outcome assessment. Some data have been excluded of the analysis for (i) lack of LID expression in the levodopa-treated condition, (ii) for misplacing of the electrodes (see section "In vivo freely moving chronic recordings" in "Methods"), and (iii) for off-target injection (see section "Neuroanatomical tracing" in Methods). Source data are provided as a Source Data.

**Reporting summary**. Further information on research design is available in the Nature Research Reporting Summary linked to this article.

## Data availability
The "data-LID" data generated in this study have been deposited in the https://www.opendata.bio.ens.psl.eu/data-LID/ database under accession LID_code_paper [https://github.com/teamnbc/LID_code_paper]. The "data-LID" data are available at https://www.opendata.bio.ens.psl.eu/data-LID/. The "data-LID" data generated in this study are provided in the Supplementary Information/Source Data file. For additional information, please refer to the corresponding Life Sciences Reporting Summary.

## Code availability
The code for electrophysiological analysis is available on Github https://github.com/teamnbc/LID_code_paper and allows verification, reproduction, interpretation, and analysis of all the figures and the supplementary figures in this study.

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

## Acknowledgements

This work was supported by Agence Nationale de Recherche to D.P (ANR-12-JSV4-0004 Ceredystim, ANR-16-CE37-0003-02 Amedyst, ANR-19-CE37-0007-01 Multimod) and to C.L. (ANR-17-CE37-0009 Mopla, ANR-17-CE16-0019 Synpredict, FRM-EQU202103012770) and to B.C. (France Parkinson, Labex Memolife) and to L.V. (Fondation Bettencourt Schueller) and by the Institut National de la Santé et de la Recherche Médicale (France). We are grateful to Sabine Meunier for the careful reading of the manuscript. We thank the Imaging Facility at IBENS (IMACHEM-IBiSA, France-BioImaging ANR-10-INBS-04, FRC Rotary International France, Investments for the future, ANR-10-LABX-54 MEMOLIFE).

## Author contributions

D.P. and C.L. acquired funding; D.P. and C.L. conceived and designed all experiments and analysis, except all patch clamp experiments designed by B.D. and L.V.; B.C., D.P., C.L. wrote the manuscript; B.C., J.L.F., E.P., A.C., T.T., C.L. analyzed the data; B.C., J.L.F., E.P., A.C., F.M., C.M.H., S.P., D.P. performed the experiments. All authors interpreted results, revised the final manuscript, and approved the final manuscript.

## Competing interests

The authors declare no competing interests.
