## [Peer Review File · Nature Communications]

Reviewers' Comments:

Reviewer #1:

Remarks to the Author:

The paper by Coutant and colleagues uses a mouse model to explore the mechanisms by which cerebellar stimulations inhibit Levodopa-induced dyskinesia (LID). LID is a common problem in treatment of Parkinson's disease. It has been demonstrated that stimulation of the cerebellar nuclei (CN) can improve symptoms for patients (e.g. Koch et al., 2009). The mechanism behind this treatment is not understood and is clearly important to explore. The novelty of this manuscript therefore lies in application of electrophysiological recording and pharmacogenetics to reveal how cerebellar stimulation alters brain activity to alleviate LIDs. However, at present, the manuscript does not provide a solid picture of the key neural changes underlying efficacy of cerebellar stimulation.

General comments:

The effect of the cerebellar stimulation on incidence of LIDs is impressive. It seems that the stimulations work with almost full efficacy on the first session (at least on oral LIDs), this makes the distinction between 'corrective' and 'preventative' appear unnecessary. Is this a fair assumption to make? If so, is it always necessary to include the preventative dataset as it greatly adds to cluttered data figures and makes it difficult to pull out salient information especially in Figures 2 and 3.

How important is the relevant timing of the stimulation and the levodopa administration? It seems that behavioural improvements outlast the stimulation sessions themselves (Fig 1e). The latter isn't really emphasised although it seems important. Figure S2 contains some of this information but there is a general ambiguity in the methods about the relative timing of drug, stimulation and behavioural/neurophysiological measurement (lines 545-546 and 677). Line 562 indicates LIDs were averaged per time point or per session – this is unclear.

The 6-OHDA model is quite severe in terms of outcomes and the initial description (Lundblad et al, 2004) reported a success rate of only 14% for animals that survive the surgery and show sufficient DA depletion. This information is not included in the manuscript. Presumably the 6-OHDA animals display Parkinsonian symptoms? Again this is not mentioned but it seems relevant. It would be helpful to know how many animals were excluded from the study due to insufficient DA depletion (line 677), and the extent of Parkinsonian symptoms within this cohort.

The manuscript reports changes in neural activity over the course of cerebellar stimulation sessions. More information is needed about the stability of neural recordings across and between behaviour sessions. For instance, were units included in the analysis if they could not be recorded over the entire duration of the session (90 mins)? Also did the incidence of LIDs decrease recording stability?

In general, the neural data is challenging to interpret: there are so many comparisons to be made, it is difficult to see the wood from the trees. In some cases, across-session changes in neural activity don't track changes in behaviour (i.e. LID score). This is the case for motor cortex (M1) in LID condition, where M1 activity ramps up over sessions whereas LID score stays flat. What is the interpretation?

In Figure 2d there is quite a large difference in baseline firing rate between the first sessions of LID and LID-CORR (see also Fig S5d). These differences are surprising at this stage. With different baselines, it is difficult to interpret the subsequent changes in cerebellar nuclear activity. In figure S5, how do the authors account for the across-session changes in the dentate and fastigial nuclei with cerebellar stimulation?

In Figure 3d, it appears that cerebellar stimulation has no effect on firing rates in the parafascicular (PF) thalamus (LID-CORR condition). However, the primary aim of figure 4 is to highlight the connection between CN and PF as putative mechanistic pathway for action of the

stimulation. I cannot see how line 280 'Because both DCN and PF showed a similar modulation of their firing rate,..' is supported by the neurophysiological data.

The change in plasticity in the striatum is very striking but the link to the rest of the paper is unclear.

Overall the paper contains a lot of information but isn't well synthesised and so lacks a coherent model of what is happening in the brain during/after cerebellar stimulation. At present, the strength of the CN-PF link is uncertain and the link to striatal plasticity is unclear.

A small note: please indicate the figure number on the figure itself, especially supplementary figures.

Reviewer #2:

Remarks to the Author:

In this manuscript, the authors have used the Levodopa-induced dyskinesia (LID) model of Parkinson's disease to test the role of cerebellar function in mediating the altered neuronal activity and behaviors that characterize this disease model. Specifically, they modulate the activity of cerebellar Purkinje cells and then examine how changing the activity of this locus impacts neural circuits at a distance. To achieve this goal, the study employs a combination optogenetics, in vivo electrophysiology, brain slice recordings for plasticity measurements, molecular marker analysis, and DREADDS. The data show compelling evidence that cerebellar stimulation indeed does have powerful functional interactions with cerebral cortical and thalamic regions, which ultimately control behavior in the LID model. This work has major relevance to basic brain circuitry and directly informs about therapy as well. The paper is exciting and timely, the experiments are masterful, and the results are beautifully presented. Below, I outline suggestions that I hope will help the authors improve the clarity. Otherwise, this is an outstanding and thorough piece of work.

- 1) I would suggest revising the title a little just to improve the flow grammatically. I would suggest "Cerebellar stimulation prevents Levodopa-induced dyskinesia in mice and normalizes brain activity".
- 2) Along the same lines as above, please look through the manuscript for minor grammatical issues. Some examples, use "cerebellar stimulation" rather than "cerebellar stimulations", and there are a number of places where the article "the" could be used properly.
- 3) To add specificity, line 28 of the summary should say "...in freely moving LID mice".
- 4) On Line 92, the authors state "Cerebellar stimulations could therefore directly restore striatal function in LID." To be accurate, this would in fact be indirect, as per the pathway you have described.
- 5) Line 98 should be plural PCs, not PC.
- 6) Much of the paper focusses on using the oral responses in LID as a readout. However, it is not clear to me just how robust this behavior actually is. I think it would really help the reader to expand the description of the behavior and add additional rationale/justification for why this specific aspect of the LID behavior was ideal for analysis in this study.
- 7) On line 122, the authors introduce the control, stating "...compared to non-lesioned levodopa-treated sham mice". This a very puzzling description to me. Please revise and expand your description of the control/sham.
- 8) On line 152, the authors state "...weeks, while stimulations starting concomitantly with levodopa administration prevented LID development...". But in this paradigm, did LID show up after you terminated treatment? That is, perhaps the concomitant stimulation was only able to suppress it, but not eliminate its emergence per se.
- 9) On line 155, "animals models" should read "animal models".
- 10) On line 159, "...concentrations (diphasic dyskinesia) review in..." should say "as reviewed in...".
- 11) On line 167 "effect than in humans were the severity" should be where, not were.
- 12) Throughout the manuscript, I would suggest using "cerebellar nuclei" rather than "deep cerebellar nuclei".
- 13) On line 180 the authors state "...exhibited an alternation of cessation of firing and increased firing relative...". Apologies, but I am having a very hard time understanding what you mean.

Please rephrase, it may help to build out this description a little more.

- 14) On line 183, the authors argue that "We verified that this did not reflect changes in motor activity." This is an important control, but please explain how you accomplished this.
- 15) On line 195, the authors state "The effects of PC stimulations were less clear in DN and FN...". Please explain what you mean by this. Do you mean that there was no effect, or that there was high variability in the effect from recording to recording, etc.?
- 16) On line 211 the authors state "The higher values of cv2.isi did not simply reflect increased bursting...". Please explain, do you mean that there was also an increase in overall irregularity, i.e. CV?
- 17) On line 244, the authors state "we chronically recorded neurons in the oral region of M1...". But what was the point of chronic recordings? Increased rate could be seen on a single session. Seems like the full interpretation/discussion of the data is missing.
- 18) On line 249 the authors state "The effects were more inconsistent in VAL". What do you mean? How do you interpret this?
- 19) On line 273, the authors state "cerebellar stimulations restore activities of both oral M1 and PF...". For this paradigm, I would argue that cerebellar stimulation prevented rather than restored cortex activity.
- 20) On line 294, "...and much less in other thalamic nuclei as VAL.." should say compared to the VAL? Please fix the grammar.
- 21) On line 300, the authors say "Since DCN may entrain PF". Please define what you mean by entrain in this context.
- 22) Line 302, "...by injecting retrograde CAV2-Cre-GFP in PF" should say "in the PF".
- 23) Also, for the DREADD experiment, what were the percentages of cells manipulated? This could probably be determined using the genetic labels in the injected constructs.
- 24) On line 368, the authors state "No significant asymmetry of striatal FosB/ Δ FosB expression was found in mice receiving 4 weeks of cerebellar stimulations (LID_PREV, N=10, 94.2 ± 3.4 %, Figure 7b, Table S8), which did not significantly differ from SHAM mice...". So, what is the interpretation for this finding?
- 25) On line 378, please fix the grammar in "FosB/ Δ FosB within the dorsolateral striatum, tightly linked to the development of LID."
- 26) Line 386 should be PCs, not PC.
- 27) This is a minor point, but you need to define how YFP was detected. With a GFP antibody? And the use of both terms needs to be defined as in some places they may have been used interchangeably, which can be confusing.
- 28) The TH expression in Figure 1C is a bit hard to see. Perhaps turn up the brightness a little.
- 29) In Figure 2C, it is confusing where the light stimulation timeline at the top is placed in relation to the raster plot below. It should be aligned with the paradigm below (but I don't think this was the intention) or clearly separated somehow.
- 30) The basal firing rate of the CN neurons in the sham rates seem quite low compared to what has previously been published. What do the raw traces look like? It would be nice to include some raw traces in the image panels.
- 31) For the schematics, for example 1B versus 2A, what is the different between PC + Chr2 and PC Chr2?
- 32) For Figure 4A, what does this injection spot look like on the actual tissue sections?
- 33) For Figure 4C, from these images shown, it is very difficult to appreciate that the Interposed Nucleus has the predominant projections.
- 34) The title of Figure 4 legend should be expanded. It is currently hard to understand.
- 35) Apologies if I am missing something, in Figure 5D why are Purkinje cells expressing the GFP and mCherry?
- 36) In Figure 7B boxplot, please explain the high variability in the LID condition.
- 37) In Figure 7B, for the lesioned side LID PREV, it looks like a lot of cells are actually positive, just lower level. Please explain.

Reviewer #3:

Remarks to the Author:

The manuscript entitled "Cerebellar stimulations prevent Levodopa-induced dyskinesia in mice and normalize brain activity" is the first study to report that optogenetic stimulation of Purkinje Cells

(PC) in CrusII (the orolingual region of the cerebellum) greatly attenuates both the development and the expression of L-DOPA-induced dyskinesia (LID) in a parkinsonian rodent model. In addition to the dyskinesia ratings, authors report strong effects of PC stimulation on several pathophysiological and molecular features, such as (i) neuronal activity in the deep cerebellar nuclei (interpositus nucleus, IN; dentate nucleus, DN; fastigial nucleus, FN); (ii) neuronal activity in the oral motor cortex; (iii) neuronal activity in the parafascicular thalamic nucleus (Pf thal); (iii) corticostriatal synaptic plasticity in direct-pathway and indirect-pathway striatal projection neurons (SPN); (iv) striatal expression of Δ FosB immunoreactivity. The results from all of these assays show that PC stimulation has a strong modulatory effect on brain networks involved in LID. Moreover, as additional investigations, the manuscript reports results from retrograde viral tracing to determine the distribution of projections from deep cerebellar nuclei (DCN) to Pf thal, and furthermore examines the effects of chemogenetically suppressing Pf thal-DCN efferent neurons to block the antidyskinetic effect of PC stimulation (finding positive results).

This is a very ambitious study reporting quite novel and interesting results. However, it is also quite laborious to read and difficult to digest. In the large amount of experiments here presented, some methodological details get lost, and the mechanistic connections between different types of results remain elusive. For example, it is very difficult to grasp how PC stimulation can mitigate the L-DOPA-induced Δ FosB upregulation in striatum. Δ FosB is a long-lived protein that accumulates in SPNs over repeated L-DOPA administration, and it remains significantly upregulated for at least 2 weeks after cessation of L-DOPA treatment, see PMID 12581184. It is also difficult to grasp how PC stimulation can modify the pattern of corticostriatal synaptic plasticity induced by L-DOPA, which is known to be strongly under the control of DA (D1 and D2) receptor stimulation on SPNs (and it is unlikely that PC stimulation modifies the levels of L-DOPA-derived DA in the striatum).

My recommendation would be to trim down and reformat the manuscript focusing on the most clearly connected parts, which are those providing evidence that LID is "a network disorder involving abnormal signaling between the basal ganglia, cerebral cortex, thalamus and cerebellum". This is indeed the most important and most novel contribution of this study, and it is likely to have a major and long-lasting impact on the field.

Some specific comments:

I. Methods

1. Specify on which day of the week dyskinesia test was performed. When presenting the data, authors indicate the week of treatment, but L-DOPA injections were performed daily. On which day of the week was dyskinesia rated?
2. It is not clear how large a part of the striatum/how many sections/animal were considered for both TH and FosB/ Δ FosB analysis.

II. Results

Figure 1:

1. Specify the dose of L-DOPA. In the methods, authors mention two doses of L-DOPA, 3 and 6 mg/kg; however, it is not clear which of the two doses was finally used for each group and for each dyskinesia test. At line 478 in the methods session, authors state that L-DOPA 3 mg/kg was administered for only 3 days and then switched to L-DOPA 6 mg/kg. However, in the results session, line 147, authors only report the dose of L-DOPA 3 mg/kg. Adding the L-DOPA dose in figures and figure legends will help the reader at better understanding the experimental design and the results.
2. The high variability in the response to L-DOPA at baseline (no optogenetic stimulation) may reflect high variability of the lesion success. Could it be that the magnitude of response to PC stimulation also depends on the extent of striatal DA denervation?

3. Authors claim that corrective optogenetic stimulation (LID_CORR) of PC in the orolingual CrusII induces a significant reduction in the orolingual AIMs and that this reduction is more pronounced than the one observed in the axial and limb dyskinesia components (line 143 and 144). However, results in Fig 1e-g indicate that the preferential effect on the orolingual component regards the preventive stimulation paradigm (LID_PREV), as the corrective stimulation produced significant decrease in all the three AIMs components compared to the LID condition.

4. I wonder whether authors have looked at the effect of the optogenetic stimulation in 6-OHDA lesioned mice in the absence of L-DOPA. Did the optogenetic stimulation of Purkinje cells affect the motor behavior? This aspect is particularly important to understand whether the optogenetic stimulation effect is selective for the AIM scores or may depend on generic motor suppression.

Figure 2:

1. Considering the low n per group (3/4) represented in this set of experiments, it is important to know how reliable the lesion was. Please report results for TH analysis as supplemental data.

Figure 3:

1. Did authors also look at the CV2.isi rather than just the firing rate?

Figure 5:

1. Was any verification made that the hM4Di DREADD actually inhibited DCN neuronal firing?

Figure 7:

1. What animals were used for the analysis of Δ FosB? Were these the same animals used in Figure 1? If yes, why does the n per group differ from that presented in Figure 1? Please specify why and if animals were excluded from the analysis. If this is a new batch of animals, please report results for the TH analysis as supplemental data.

2. Was the person performing Δ FosB cell counts blind to animal ID or experimental groups?

Reviewer #4:

Remarks to the Author:

This study demonstrates a remarkable influence of cerebellar neurostimulation on levodopa induced dyskinesia (LID) and striatal plasticity in a rodent model of Parkinson's disease. Daily stimulation via optogenetics of Purkinje cells in the orolingual region of the cerebellum evoked a surprisingly long-lasting reduction in oromotor LID with concomitant restoration of aberrant firing rates in motor cortex and thalamus, a reduction in LID induced striatal FOS expression, and a marked alteration of synaptic plasticity in striatal neurons that express D1 receptors. These findings are important — they reveal that cerebellar neurostimulation can provide a therapeutic means for improving dysfunctional physiological signaling across distributed brain circuits, most notably via the thalamic parafascicularis influence on dorsal striatum. The main weakness of the paper in its current form is that insufficient numbers of cerebellar nucleus neurons were recorded to make robust conclusions about population firing rates. The phrase "normalize brain activity" in the title overstates the findings, in which extracellular recordings subject to sampling biases were restricted to a subset of brain regions. Some of the figures are not easily interpretable. With appropriate revisions, this study provides a fundamental and exciting advance in our understanding of circuit mechanisms that could be leveraged to ameliorate dyskinesia in Parkinson's and other disorders linked with striatal dysfunction.

Major

1. High variability in DCN firing rates across control conditions preclude any firm conclusions about restoration of normal firing. The sample sizes for DCN are in the tens or fewer, vs hundreds for motor cortex and PF. Many more units need to be recorded from DCN given heterogeneity of firing

rates. The CV2 result looks interesting, but it's hard to interpret without seeing the corresponding firing patterns. Given challenges with recording stability in freely moving animals, were only largest neurons likely to be recorded? Results from motor cortex look more robust, but it's not clear whether recordings were restricted to pyramidal cells or also included interneurons.

2. Citations about previous demonstrations of synaptic connections from DCN to parafasciculus and disynaptically to striatum are missing from the introduction and results and mentioned in passing in the Discussion (references 83-85). Figure 4 appears to be a simple replication of previous findings on DCN to PF projections in mice, with no new information that would justify inclusion in this paper.

Figures

1. General - figures with bar graphs have unexplained color dots which do not appear to correspond with the box plot values. The meaning of the box plots needs to be clarified in the figure legends. The many figures with multiple sequential box plots are difficult to parse - they seem cluttered with lines and dots but don't have an indication of the specific time window for each box plot.

2. Figure 2. It looks like only a subset of PCs are labeled- is this an artifact of contrast adjustment?

2. Figure 5 - the mCherry and GFP signals are not adequately bright.

3. Figure 7 - the FOS staining is not adequately bright.

Minor

1. line 60 - please clarify the species here

2. line 63+ The motivation for the experiments needs to be made more explicit. "A first possibility" - for what?

3. line 73. relay to where? the fundamental circuit needs to be clarified here.

4. line 91. add references 84, 85 and note that PF is considered part of IL thalamus.

5. line 117. the classic model needs a reference

6. line 149 replace 'none' with 'no'

7. line 167, 'were' - incorrect word

8. lines 302. remove 'therefore'

9. line 575. provide specific parameters for theta stimulation.

10. line 594 - missing s (s)pontaneous

11. lines 632-3. Add controls for off-target injections and verification of injection sites?

Responses to reviewers

We wish to thank you for considering our revised manuscript on the role of cerebellar stimulation on levodopa-induced dyskinesia in mice for publication in Nature Communications.

We are extremely grateful to our colleagues for their careful reading of the manuscript and for their very constructive comments. We did our best to improve the manuscript under their guidance and we feel the manuscript has immensely benefited from their thoughtful suggestions and requests. We made numerous modifications and clarifications in the manuscript and in the figures, and also performed a number of additional analysis and experiments, which delayed our resubmission. We believe that the modifications, and the additional analysis and experiments that we performed, have significantly improved the readability of our findings and the strength of our conclusions. We hope that our work is now ready for publication.

Please find below a point by point response to the reviewers comments.

Reviewer #1 (Remarks to the Author):

The paper by Coutant and colleagues uses a mouse model to explore the mechanisms by which cerebellar stimulations inhibit Levodopa-induced dyskinesia (LID). LID is a common problem in treatment of Parkinson's disease. It has been demonstrated that stimulation of the cerebellar nuclei (CN) can improve symptoms for patients (e.g. Koch et al., 2009). The mechanism behind this treatment is not understood and is clearly important to explore. The novelty of this manuscript therefore lies in application of electrophysiological recording and pharmacogenetics to reveal how cerebellar stimulation alters brain activity to alleviate LIDs. However, at present, the manuscript does not provide a solid picture of the key neural changes underlying efficacy of cerebellar stimulation.

General comments:

The effect of the cerebellar stimulation on incidence of LIDs is impressive. It seems that the stimulations work with almost full efficacy on the first session (at least on oral LIDs), this makes the distinction between "corrective" and "preventative" appear unnecessary. Is this a fair assumption to make? If so, is it always necessary to include the preventative dataset as it greatly adds to cluttered data figures and makes it difficult to pull out salient information especially in Figures 2 and 3.

This is a good remark and would indeed unclutter the figures if we could merge these two groups. However, the distinction between “corrective” and “preventive” animals remains necessary as corrective animals first developed LID, with the associated anomalies in motor circuits described in other studies. In preventive animals, the stimulations are performed before the development of LID, with therefore another state of the motor circuits. The anomalies associated to LID are not yet into place and the state of the motor circuits is probably not the same as corrective mice, rendering the merge of these two groups impossible. Lines 188-190 have been added to emphasize this important concern. Moreover, we realized that the days on which LID are scored are very unclear (and actually the observed effects are not on the first session of cerebellar stimulation). Thank you for rising that important concern. A detailed schematic of the days of LID recordings during the protocol in Figure 1 has been added (Figure 1a – bottom) and lines 134-136 emphasized the exact time points where LID have been scored. Moreover, corrective mice have been removed from Figures 2, 3, and 7 in order to uncluttered the major findings of our study. Indeed, corrective animals were first necessary to reproduced the effects observed in Koch et al. 2009 on humans (Figure 1c-g) and to show he involvement of Purkinje cells. However, as the beneficial effects of cerebellar stimulation can even prevent the appearance of LID, preventive animals represent a stronger and more interesting condition for the alleviation of LID (Lines 214-218). We do hope our paper will be clearer with this removal.

How important is the relevant timing of the stimulation and the levodopa administration? It seems that behavioural improvements outlast the stimulation sessions themselves (Fig 1e). The latter isn't really emphasised although it seems important. Figure S2 contains some of this information but there is a general ambiguity in the methods about the relative timing of drug, stimulation and behavioural/neurophysiological measurement (lines 545-546 and 677). Line 562 indicates LIDs were averaged per time point or per session' this is unclear.

Thank you for this remark. Indeed, the information about the relative timing of drug, cerebellar stimulation, and behavioral/electrophysiological measurements was unclear. The timing of stimulation compared to the levodopa administration has been clarified in lines 157-159 in the Results section. Moreover, in the Methods section, lines 607-610 were added to limit this ambiguity as well as a new scheme in Figure 1 (Figure1 – bottom). Lines 163-164, 195-196, and 412-415, 444, and 467 in the “Results” and lines 552-556 and lines 574-575 in “Discussion” sections emphasize the behavioral improvements outlasting the stimulation.

Line 562 (now 693-694) has been rephrased.

The 6-OHDA model is quite severe in terms of outcomes and the initial description (Lundblad et al, 2004) reported a success rate of only 14% for animals that survive the surgery and show sufficient DA depletion. This information is not included in the manuscript. Presumably the 6-OHDA animals display Parkinsonian symptoms? Again this is not mentioned but it seems relevant. It would be helpful to know how many animals were excluded from the study due to insufficient DA depletion (line 677), and the extent of Parkinsonian symptoms within this cohort.

The post-operative survival rate has been added in the “Methods” section, line 599. All condition included, our survival rate reached 80%. We have a survival rate of 50% in parkinsonian mice, between the one observed in (Lundblad *et al.*, 2004) – 14% and the one observed in (Francardo *et al.*, 2011) – 100% with refined protocols of animal nursing as, in our case, in addition to the MFB lesion, animals were implanted with at least three cannulas of electrodes in several regions of the brain. The entire surgery lasts longer and is therefore heavier for the animal to recover. 6-OHDA-lesioned mice indeed presented parkinsonian symptoms, we analyzed them and a supplementary figure showing the bias of PD mice in ipsilateral use of their forepaw (Supplementary Figure S1) has been added in order to highlight the parkinsonian symptoms presented by 6-OHDA-lesioned animals. Moreover, Menardy *et al.* 2019 from our lab characterized parkinsonian symptoms following 6-OHDA MFB depletion in this mouse line. Lines 127-129 have been added in order to clarify the missing information. No animal has been excluded from the study due to insufficient DA depletion explaining the 50% threshold. The sentence (former line 677) has been rephrased as it was indeed unclear (now lines 831-836).

The manuscript reports changes in neural activity over the course of cerebellar stimulation sessions. More information is needed about the stability of neural recordings across and between behaviour sessions. For instance, were units included in the analysis if they could not be recorded over the entire duration of the session (90 mins)? Also did the incidence of LIDs decrease recording stability?

Thank you for this question. Across one session, we spike-sorted recorded units over the entire recording time (90min) therefore the spike sorted units are present over the entire recording session. Lines 738-740 were added in order to address this concern. Across sessions, no statement can be made on whether the units are the same across the different sessions (lines 742-744) is made. However, in the boxplots (Figures 2 and 3), only a 10min recording window corresponding to the peak of L-DOPA effect and right after the cerebellar stimulation is shown as averaging the entire recording with changes in L-DOPA concentrations and cerebellar stimulation occurring at t=30min post-levodopa injection would average different modulations of neural activity. Lines 744-749 have been added in order to clarify this concern. The incidence of LID did not decrease the recording stability more than the usual difficulty to record freely moving animals.

In general, the neural data is challenging to interpret: there are so many comparisons to be made, it is difficult to see the wood from the trees. In some cases, across-session changes in neural activity don't track changes in behaviour (i.e. LID score). This is the case for motor cortex (M1) in LID condition, where M1 activity ramps up over sessions whereas LID score stays flat. What is the interpretation?

Thank you for this excellent question. We believed that the ramping activity in M1 is secondary to LID development in dyskinetic mice. The exogenous action of L-DOPA on dopamine receptors within the striatum is immediate due to high L-DOPA dosage and hypersensitive D1R in striatonigral neurons, leading to the immediate emergence of abnormal movements.

However, the global activity of M1 gradually ramping might reflect slower mechanisms such as changes in plasticity and/or connectivity which might not be primarily involved in the generation of the abnormal movements but gradually locks the motor circuits in this pathological state. Lines 511-517 in the Discussion section have been added in order to discuss this important observation.

Concerning the cluttered figures, all electrophysiological figures were changed and focus now in the effects observed in preventive animals as 1) the preventive effect is stronger in terms of relevance for the treatment of LID and 2) electrophysiological results in corrective animals are more variable, probably due to the fact that motor circuits go through many different states (dyskinetic, slower decrease in LID severity - correction, long-term corrections) which is more difficult to address in only a few time points.

In Figure 2d there is quite a large difference in baseline firing rate between the first sessions of LID and LID-CORR (see also Fig S5d). These differences are surprising at this stage. With different baselines, it is difficult to interpret the subsequent changes in cerebellar nuclear activity. In figure S5, how do the authors account for the across-session changes in the dentate and fastigial nuclei with cerebellar stimulation?

That's indeed an important observation. Deep cerebellar nuclei are very heterogeneous structures with many different neuronal populations and a wide range of firing rate *in vivo* (0-100Hz) (Canto, Witter and Zeeuw, 2016). These features might explain the high variability observed between groups in the "Pre-Levodopa" phase and that's why we decided to make comparisons inside the groups themselves rather than between conditions as indeed the subsequent changes are difficult to interpret (lines 232-235). Moreover, we failed to show any consistent modulations in DN and FN intra-group, compared to IN, either due to the low number of cells and high variability or because these two nuclei are less recruited in both the dyskinetic and the preventive states.

In Figure 3d, it appears that cerebellar stimulation has no effect on firing rates in the parafascicular (PF) thalamus (LID-CORR condition). However, the primary aim of figure 4 is to highlight the connection between CN and PF as putative mechanistic pathway for action of the stimulation. I cannot see how line 280 "Because both DCN and PF showed a similar modulation of their firing rate..."; is supported by the neurophysiological data.

The sentence has been rephrased (lines 353-357). It was indeed unclear. We meant that CN and PF are similarly modulated during and following cerebellar stimulation in preventive mice and following LDOPA treatment in dyskinetic mice. The effects of cerebellar stimulation are indeed harder to identify in corrective mice probably due to the many transitions that the motor systems have to go through, which is avoided in preventive mice as cerebellar stimulation started before the changes associated to the dyskinetic state are established.

The change in plasticity in the striatum is very striking but the link to the rest of the paper is unclear.

Aberrant LTP at corticostriatal synapses has been tightly linked to LID expression (Picconi *et al.*, 2003, 2004; Thiele *et al.*, 2014; Calabresi *et al.*, 2016). As cerebellar stimulation prevents LID development in preventive animals, it was necessary to study how cerebellar stimulation might affect LID hallmarks such as Δ FosB and aberrant LTP. Moreover, as beneficial effects outlast cerebellar stimulation, it was important to know how this was translated into striatal long-term changes. Lines 412-415 in the Results section and lines 549-556 in the Discussion section were modified and emphasized the reasons why we thought this experience was of great importance. Moreover, Figure 8 has been added for a better visualization of the potential effects of cerebellar stimulation on the circuitry.

An interpretation of a link between the change in striatal plasticity and cerebellar stimulation is indeed greatly missing from this manuscript. Therefore, a putative link between cerebellar stimulation, Δ FosB normalization, and LTD in striatonigral neurons has been added in the "Discussion" section (lines 569-574).

Overall the paper contains a lot of information but isn't well synthesised and so lacks a coherent model of what is happening in the brain during/after cerebellar stimulation. At present, the strength of the CN-PF link is uncertain and the link to striatal plasticity is unclear.

The link between cerebellar stimulation and striatal plasticity was indeed missing from the manuscript. Lines 549-556 and lines 568-573 in the Discussion section were added in order to address this concern. Some parts in the "Discussion" section have been more developed, especially about the putative link between cerebellar stimulation, Δ FosB, and corticostriatal plasticity (lines 568-573). Moreover, Figure 8 has been added with the hope to better clarify what might be going on in the upper motor circuits before and after cerebellar stimulation in dyskinesia.

A small note: please indicate the figure number on the figure itself, especially supplementary figures.

This has been added.

Reviewer #2 (Remarks to the Author):

In this manuscript, the authors have used the Levodopa-induced dyskinesia (LID) model of Parkinson's disease to test the role of cerebellar function in mediating the altered neuronal activity and behaviors that characterize this disease model. Specifically, they modulate the activity of cerebellar Purkinje cells and then examine how changing the activity of this locus

impacts neural circuits at a distance. To achieve this goal, the study employs a combination of optogenetics, in vivo electrophysiology, brain slice recordings for plasticity measurements, molecular marker analysis, and DREADDS. The data show compelling evidence that cerebellar stimulation indeed does have powerful functional interactions with cerebral cortical and thalamic regions, which ultimately control behavior in the LID model. This work has major relevance to basic brain circuitry and directly informs about therapy as well. The paper is exciting and timely, the experiments are masterful, and the results are beautifully presented. Below, I outline suggestions that I hope will help the authors improve the clarity. Otherwise, this is an outstanding and thorough piece of work.

1) I would suggest revising the title a little just to improve the flow grammatically. I would suggest “Cerebellar stimulation prevents Levodopa-induced dyskinesia in mice and normalizes brain activity”.

Done. Sorry for the many grammar mistakes and thank you.

2) Along the same lines as above, please look through the manuscript for minor grammatical issues. Some examples, use “cerebellar stimulation” rather than “cerebellar stimulations”, and there are a number of places where the article “the” could be used properly.

We tried to do our best to correct any grammatical errors we could observe. We hope this version will be better grammatically speaking.

3) To add specificity, line 28 of the summary should say “in freely moving LID mice”.

Because preventive mice are not really LID mice, we preferred to stay more general, however it has been changed. Thank you.

4) On Line 92, the authors state “Cerebellar stimulations could therefore directly restore striatal function in LID”. To be accurate, this would in fact be indirect, as per the pathway you have described.

That’s right. A more specific sentence has been added.

5) Line 98 should be plural PCs, not PC.

Changed. Thank you.

6) Much of the paper focusses on using the oral responses in LID as a readout. However, it is not clear to me just how robust this behavior actually is. I think it would really help the reader to expand the description of the behavior and add additional rationale/justification for why this specific aspect of the LID behavior was ideal for analysis in this study.

That is an excellent question. In Methods, lines 686-689 were added to describe in more details abnormal oral behavior considered as oral LID. Oral LID represents a powerful readout of the potential beneficial impacts of cerebellar stimulation, centered on CrusII, on the pathological phenotype as LID can be divided into different subtypes: oral, limb, and axial LID and the fractured somatotopy of the cerebellum has described a specific region in the cerebellar hemispheres, CrusII, hosting dense projections from the orolingual area (Apps et al. 2009). Lines 129-134 were added to address this important point.

7) On line 122, the authors introduce the control, stating “compared to non-lesioned levodopa-treated sham mice”. This a very puzzling description to me. Please revise and expand your description of the control/sham.

The sentence was indeed unclear. Control mice are lesioned with a vehicle (so non-(6-OHDA-) lesioned animals) and treated with L-DOPA as LID mice. They are also called SHAM mice. Lines 138-140 have been rephrased.

8) On line 152, the authors state “weeks, while stimulations starting concomitantly with levodopa administration prevented LID development”. But in this paradigm, did LID show up after you terminated treatment? That is, perhaps the concomitant stimulation was only able to suppress it, but not eliminate its emergence per se.

Thank you for this excellent remark. The beneficial effects on oral LID outlast the cerebellar stimulation (week 8), however, at week 9, some animals showed a small resurgence of oral LID in preventive animals (See Figure 1e). The severity of oral LID in preventive mice is still significantly different than dyskinetic animals but became also different than the severity observed in SHAM animals. We did not look after the 9th week. Lines 163-164 and lines 195-196 in the Results Section and lines 555-557 and lines 576-578 in the Discussion Section have been edited or added to emphasize the long-lasting effect of cerebellar stimulation.

Moreover, as preventive cerebellar stimulation acts on long-term changes associated with LID such as transcription factor expression (FosB) and plasticity at corticostriatal synapses and both experiences were performed right after week 9, it might suggest the effects of preventive cerebellar stimulation acts on profound mechanisms associated to LID, thus allowing long-term alleviation. A longer experimental protocol (> 10 weeks) would be necessary to answer that question. Lines 569-574 in the Discussion Section were added to better address this concern.

9) On line 155, “animals models” should read “animal models”.

Changed (line 201). Thank you.

10) On line 159, “concentrations (diphasic dyskinesia) review in” should say “as reviewed in”.

Changed (line 205). Thank you.

11) On line 167 “effect than in humans were the severity” should be where, not were.

Changed (line 213). Thank you.

12) Throughout the manuscript, I would suggest using “cerebellar nuclei” rather than “deep cerebellar nuclei”.

As cerebellar nuclei might include vestibular nuclei, we choose “deep cerebellar nuclei” to avoid any confusion. I changed it though. Thank you.

13) On line 180 the authors state “exhibited an alternation of cessation of firing and increased firing relative”. Apologies, but I am having a very hard time understanding what you mean. Please rephrase, it may help to build out this description a little more.

This sentence was indeed unclear and has been rephrased for better clarity (lines 228-232). Thank you.

14) On line 183, the authors argue that “We verified that this did not reflect changes in motor activity”. This is an important control, but please explain how you accomplished this.

Lines 238-239 have been added for better clarity. A more complete explanation of how we performed this analysis is present in a supplementary text to unclutter the major findings. To verify that the modulations of CN activity was not due to changes in locomotor activity, we quantified locomotor activity in control mice using DeepLabCut. No differences were observed in CN activity during levodopa treatment in control mice, whether the animals were moving or not (Supplementary Figures 7 and 8).

15) On line 195, the authors state “The effects of PC stimulations were less clear in DN and FN”. Please explain what you mean by this. Do you mean that there was no effect, or that there was high variability in the effect from recording to recording, etc.?

The sentence has been rephrased (lines 253-256). We meant that we failed to show any consistent modulations following cerebellar stimulations in the DN and the FN of corrective and preventive animals. This lack of effects might be caused by the fact that they are less recruited in the alleviation of the dyskinetic phenotype. Indeed, in addition to the consistent modulations observed in the IN following cerebellar stimulation, its dense projections to both PF and Crus II point toward IN as a prime candidate in the alleviation of LID.

16) On line 211 the authors state “The higher values of cv2.isi did not simply reflect increased bursting”. Please explain, do you mean that there was also an increase in overall irregularity, i.e. CV?

This sentence was indeed unclear and has been modified in order to be more explicit (lines 272-273). We meant that since the increase in cv2.isi observed in LID mice may have reflected increase burst rate, we analyzed the burst rate during periods of locomotor activity in LID mice. Interestingly, the burst rate of LID mice decreased during levodopa treatment in the IN whereas cv2.isi increased. Therefore, the observed increase in cv2.isi in the IN of LID mice does not reflect increase burst rate but rather, might reflect increased pause following levodopa treatment. Moreover, cv.isi were added in Supplementary Figure S6 in CN in the different conditions but no significant modulations could be observed.

17) On line 244, the authors state “we chronically recorded neurons in the oral region of M1”. But what was the point of chronic recordings? Increased rate could be seen on a single session. Seems like the full interpretation/discussion of the data is missing.

In LID mice indeed, the increased rate could be seen on a single session, however, chronic recordings were necessary to study the evolution of the firing rate through the entire protocol, especially preventive animals. Moreover, we continued to record activities after the cerebellar stimulation ended in order to determine if the modulations of the firing rate during cerebellar stimulation were persistent or only generated by the concomitance of levodopa administration and cerebellar stimulation. Lines 309-311 were added in order to clarify the necessity in this protocol to use chronic recordings.

18) On line 249 the authors state “The effects were more inconsistent in VAL”. What do you mean? How do you interpret this?

Line 249 was indeed unclear and has been changed for better clarity (Lines 316-318 and lines 337-339). We meant that we failed to show any consistent modulations in VAL activity in both dyskinetic animals and in mice receiving cerebellar stimulations. We recorded in the cerebellar-recipient region of VAL (information that was missing from the manuscript and has been added in line 308) and expected to see modulations associated to LID. However, either the modulations of VAL are more subtle or the cerebellar-territory of this region is not as recruited as the PF in both LID and LID alleviation. Some authors working on dyskinetic patients have indeed suggested that cerebellar stimulation using rTMS might act primarily on the sensory afferent volley reaching M1 rather than direct effects on the motor afferents through VAL (Popa *et al.*, 2013). This hypothesis has been added to the Discussion in lines 530-533 and lines 544-545. Moreover, lesions of the basal ganglia-recipient region of VAL but not its cerebellar-territory, reduced LID in non-human primates, suggesting a distinct role of these two subregions of VAL in LID (Page, Sambrook and Man, 1993).

19) On line 273, the authors state “cerebellar stimulations restore activities of both oral M1 and PF”. For this paradigm, I would argue that cerebellar stimulation prevented rather than restored cortex activity.

Right. This has been changed.

20) On line 294, “and much less in other thalamic nuclei as VAL..” should say compared to the VAL? Please fix the grammar.

The sentence has been rephrased as “with very little collaterals to VAL” (now lines 370-371), as the goal of this experiment was to determine if VAL might have been involved in the effects of the inhibition of the CN-PF pathway on the oral dyskinetic behavior through collaterals, which is not the case. Sorry for the grammar.

21) On line 300, the authors say “Since DCN may entrain PF”. Please define what you mean by entrain in this context.

Sentence rephrased (lines 379-380). We meant that CN could modulate the activity of PF as both present similar modulations following PC stimulations and CN project massively to the PF.

22) Line 302, “by injecting retrograde CAV2-Cre-GFP in PF”; should say “in the PF”.

Changed. Thank you.

23) Also, for the DREADD experiment, what were the percentages of cells manipulated? This could probably be determined using the genetic labels in the injected constructs.

Thank you for this excellent question. We performed this analysis and found that 22.89% of the CN cells projecting to PF were expressing DREADDs, which would assure proper inhibition of the pathway. This analysis has been added to Figure 5 (Figure 5e) and in the text (Lines 390-393 in the Results Section and lines 793-795 in the Methods Section).

24) On line 368, the authors state “No significant asymmetry of striatal FosB/ Δ FosB expression was found in mice receiving 4 weeks of cerebellar stimulations (LID_PREV, N=10, 94.2 +/- 3.4 %, Figure7b, Table S8), which did not significantly differ from SHAM mice”. So, what is the interpretation for this finding?

Lines 461-463 were added in order to better clarify what these results suggest. Moreover, lines 568-573 in the “Discussion” section speculate on what could be the putative mechanisms explaining this effect. We believed that cerebellar stimulation promotes LTD at corticostriatal synapses of D1-MSNs through the regulation of transcription factor involved in D1 receptor

signaling pathway and LTP/LTD such as ERK1/2. ERK1/2 is upregulated in LID and is involved in the regulation of the transcription of FosB/ Δ FosB. Therefore, the downregulation of ERK1/2 by cerebellar stimulation through the promotion of LTD could in turn downregulate FosB/ Δ Fos, leading to LID alleviation. Further experiments remain necessary to confirm the hypothesis.

25) On line 378, please fix the grammar in “FosB/ Δ FosB within the dorsolateral striatum, tightly linked to the development of LID”.

We are not sure what is the grammar mistake present in that sentence but a verb has been added (now line 471).

26) Line 386 should be PCs, not PC.

Changed. Thank you.

27) This is a minor point, but you need to define how YFP was detected. With a GFP antibody? And the use of both terms needs to be defined as in some places they may have been used interchangeably, which can be confusing.

Both YFP (expressed in PCs in L7-ChR2-YFP mouse line) and GFP (present in viruses used for the viral tracing and chemogenetic experiments) were detected using a confocal microscope with filters adapted to the emission light of the fluorescent protein. No antibodies were necessary to detect fluorescent signal. This has been added in the “Methods” section lines 803-806.

YFP has been defined in line 101; GFP has been defined in lines 367-368.

The text has been screened for any potential misuses of either YFP and GFP. Thank you.

28) The TH expression in Figure 1C is a bit hard to see. Perhaps turn up the brightness a little.

The image showing the TH expression in Figure 1c has been retaken for better visualization. Thank you.

29) In Figure 2C, it is confusing where the light stimulation timeline at the top is placed in relation to the raster plot below. It should be aligned with the paradigm below (but I don't think this was the intention) or clearly separated somehow.

The alignment was indeed confusing. Figure 2C has been corrected. Thank you.

30) The basal firing rate of the CN neurons in the sham rates seem quite low compared to

what has previously been published. What do the raw traces look like? It would be nice to include some raw traces in the image panels.

Raw traces in SHAM animals have been added in Supplemental Figure 8.

31) For the schematics, for example 1B versus 2A, what is the different between PC + Chr2 and PC Chr2?

There are no differences, it was a mistake. It has been changed. Thank you.

32) For Figure 4A, what does this injection spot look like on the actual tissue sections?

The injection spot in PF has been in Figure 4 (Figure 4a – inset).

33) For Figure 4C, from these images shown, it is very difficult to appreciate that the Interposed Nucleus has the predominant projections.

Indeed. The images have been changed in order to clarify this important point and for better appreciation of the prominent projections from the IN to PF. Thank you.

34) The title of Figure 4 legend should be expanded. It is currently hard to understand.

The main point of Figure 4 was indeed unclear and its title has been changed.

35) Apologies if I am missing something, in Figure 5D why are Purkinje cells expressing the GFP and mCherry?

That is an excellent remark. The image has been taken with light emissions' filters overlapping. This has been corrected and the image retaken.

36) In Figure 7B boxplot, please explain the high variability in the LID condition.

That's a good question. One section / animal in the dorsolateral part of the middle striatum (~ AP: 0.50 mm; DV: -3.0 mm; ML: +2.5 mm) was used for quantification (Methods: line 822-823). However, LID mice express different LID subtypes in different proportions and the different LID subtypes present a topographical organization in the expression of FosB/ Δ FosB (ref: Cenci et al. 1999) which could account for the variability observed in the LID group.

37) In Figure 7B, for the lesioned side LID PREV, it looks like a lot of cells are actually positive, just lower level. Please explain.

Preventive and control animals do present a basal expression of FosB/ Δ FosB (added in line 450) which explain why positive cells are observed and why we compared the unlesioned striatum with the lesioned one. However, the images were not very clear and have been retaken for better visualization. Thank you.

Reviewer #3 (Remarks to the Author):

The manuscript entitled “Cerebellar stimulations prevent Levodopa-induced dyskinesia in mice and normalize brain activity” is the first study to report that optogenetic stimulation of Purkinje Cells (PC) in CrusII (the orolingual region of the cerebellum) greatly attenuates both the development and the expression of L-DOPA-induced dyskinesia (LID) in a parkinsonian rodent model. In addition to the dyskinesia ratings, authors report strong effects of PC stimulation on several pathophysiological and molecular features, such as (i) neuronal activity in the deep cerebellar nuclei (interpositus nucleus, IN; dentate nucleus, DN; fastigial nucleus, FN); (ii) neuronal activity in the oral motor cortex; (iii) neuronal activity in the parafascicular thalamic nucleus (Pf thal); (iii) corticostriatal synaptic plasticity in direct-pathway and indirect-pathway striatal projection neurons (SPN); (iv) striatal expression of Δ FosB immunoreactivity. The results from all of these assays show that PC stimulation has a strong modulatory effect on brain networks involved in LID. Moreover, as additional investigations, the manuscript reports results from retrograde viral tracing to determine the distribution of projections from deep cerebellar nuclei (DCN) to Pf thal, and furthermore examines the effects of chemogenetically suppressing Pf thal-DCN efferent neurons to block the antidyskinetic effect of PC stimulation (finding positive results).

This is a very ambitious study reporting quite novel and interesting results. However, it is also quite laborious to read and difficult to digest. In the large amount of experiments here presented, some methodological details get lost, and the mechanistic connections between different types of results remain elusive. For example, it is very difficult to grasp how PC stimulation can mitigate the L-DOPA-induced Δ FosB upregulation in striatum. Δ FosB is a long-lived protein that accumulates in SPNs over repeated L-DOPA administration, and it remains significantly upregulated for at least 2 weeks after cessation of L-DOPA treatment, see PMID 12581184. It is also difficult to grasp how PC stimulation can modify the pattern of corticostriatal synaptic plasticity induced by L-DOPA, which is known to be strongly under the control of DA (D1 and D2) receptor stimulation on SPNs (and it is unlikely that PC stimulation modifies the levels of L-DOPA-derived DA in the striatum).

My recommendation would be to trim down and reformat the manuscript focusing on the most clearly connected parts, which are those providing evidence that LID “a network disorder involving abnormal signaling between the basal ganglia, cerebral cortex, thalamus and

cerebellum". This is indeed the most important and most novel contribution of this study, and it is likely to have a major and long-lasting impact on the field.

Some specific comments:

I. Methods

1. Specify on which day of the week dyskinesia test was performed. When presenting the data, authors indicate the week of treatment, but L-DOPA injections were performed daily. On which day of the week was dyskinesia rated?

The information about the relative timing of drug, cerebellar stimulations, and behavioral measurements was indeed unclear. In the "Methods" section, lines 607-608, lines 615-616, and lines 693-696 were added to limit this ambiguity as well as a new scheme in Figure 1 (Figure 1a – bottom) representing the day of each week during which LID were rated. Probably, one of the major information missing is that we started scoring dyskinesia on day 4 which corresponds to the first day of 6mg/kg L-DOPA concentration after three days during which all mice received 3mg/kg.

2. It is not clear how large a part of the striatum/how many sections/animal were considered for both TH and FosB/ Δ FosB analysis.

Indeed. Lines 822-823 (for FosB) and lines 828-830 (for TH) in the "Methods" section were added to specify the region of the striatum and the number of sections per animal used in the analyses of both TH and FosB/ Δ FosB. For the TH analysis, one section / animal of the middle striatum (bregma 0.50 – bregma 0.25) was used for TH quantification, even though we verified that the lesion was present in the posterior and anterior striatum as well. For FosB/ Δ FosB analysis, one section / animal in the dorsolateral part of the middle striatum (~ AP: 0.50 mm; DV: -3.0 mm; ML: +2.5 mm) was used for quantification.

II. Results

Figure 1:

1. Specify the dose of L-DOPA. In the methods, authors mention two doses of L-DOPA, 3 and 6 mg/kg; however, it is not clear which of the two doses was finally used for each group and for each dyskinesia test. At line 478 in the methods session, authors state that L-DOPA 3 mg/kg was administered for only 3 days and then switched to L-DOPA 6 mg/kg. However, in the results session, line 147, authors only report the dose of L-DOPA 3 mg/kg. Adding the L-DOPA

dose in figures and figure legends will help the reader at better understanding the experimental design and the results.

Yes, indeed that is an important point, thank you. A new scheme in Figure 1 representing the day of each week during which LID were rated and the start of the 6mg/kg dosage of L-DOPA was added for more clarity. Lines 134-136 of the "Results" section were also modified to avoid any confusion. We also added the L-DOPA dosage in the figures' legends.

2. The high variability in the response to L-DOPA at baseline (no optogenetic stimulation) may reflect high variability of the lesion success. Could it be that the magnitude of response to PC stimulation also depends on the extent of striatal DA denervation?

Thank you for this very good question. To answer it, we correlated our behavioral effect with the extent of the lesion. No correlation was observed between the extent of the DA lesion and the score of severity of oral LID in the different conditions, suggesting that the magnitude of the response to cerebellar stimulation does not depend on the severity of the DA lesion. Only SHAM animals presented a small positive correlation, which is less surprising as if some dopaminergic fibers are depleted in control conditions, it leads to the development of small abnormal movements (Table S4). We added lines 197-200 in the "Results" section describing these results.

3. Authors claim that corrective optogenetic stimulation (LID_CORR) of PC in the orolingual CrusII induces a significant reduction in the orolingual AIMs and that this reduction is more pronounced than the one observed in the axial and limb dyskinesia components (line 143 and 144). However, results in Fig 1e-g indicate that the preferential effect on the orolingual component regards the preventive stimulation paradigm (LID_PREV), as the corrective stimulation produced significant decrease in all the three AIMs components compared to the LID condition.

That is correct. We modified lines 167-168 in order to correct the confusion. We first claimed that corrective stimulation of PCs in the orolingual Crus II induced a significant reduction in the orolingual AIMs compared to limb and axial based on 1) the severity of axial LID remains different than SHAM during and after cerebellar stimulation in corrective mice; 2) the severity of limb LID was already different from dyskinetic mice in corrective mice, therefore this difference could not be attributed to PCs' stimulations. Moreover, LID mice also present a decrease in severity of limb AIMs in week 7 (third boxplot) rendering impossible to assess if the similar decrease observed in severity in limb LID in corrective animals is due to our PCs' stimulations.

4. I wonder whether authors have looked at the effect of the optogenetic stimulation in 6-OHDA lesioned mice in the absence of L-DOPA. Did the optogenetic stimulation of Purkinje cells affect the motor behavior? This aspect is particularly important to understand whether the optogenetic stimulation effect is selective for the AIM scores or may depend on generic motor suppression.

We did not stimulate PCs in 6-OHDA-lesioned animals in the absence of L-DOPA in this study. However, Menardy and colleagues from our team stimulated PCs over Crus I in L7-ChR2-YFP mouse line and observed only small head movements following cerebellar stimulation that were similar in both SHAM and 6-OHDA-lesioned mice (Menardy *et al.*, 2019). Moreover, Proville and colleagues from our team observed a small backward shift of the whiskers triggered by PCs opto-stimulations over Crus I in healthy L7-ChR2-YFP (Proville *et al.*, 2014). These results suggest that PCs' stimulation may trigger movements generation. In our study, SHAM animals treated with L-DOPA received PCs' stimulations over Crus II (SHAM_CORR and SHAM_PREV) (Figure S4). No excessive movements nor suppression of naturally occurring mouth movements were observed in control mice following PCs' stimulations. Lines 149-150 were added in order to highlight this important concern.

Figure 2:

1. Considering the low n per group (3/4) represented in this set of experiments, it is important to know how reliable the lesion was. Please report results for TH analysis as supplemental data.

That is indeed an important remark. We added TH analysis from the animals used in the experiment corresponding to Figure 2 (Figure 2b – bottom right) and found no significant differences in the extent of the dopaminergic lesion between our different conditions.

Figure 3:

1. Did authors also look at the CV2.isi rather than just the firing rate?

We did look at cv.isi and cv2.isi in the oral motor cortex, the parafascicular nucleus, and the ventroanterior-ventrolateral complex of the thalamus. However, no consistent modulations of cv2.isi have been observed in these structures following either L-DOPA treatment nor cerebellar stimulation. We added the plots corresponding to this analysis in Supplementary Figure S10e and f.

Figure 5:

1. Was any verification made that the hM4Di DREADD actually inhibited DCN neuronal firing?

We did not combine electrophysiological recordings of CN activity with the chemogenetic inhibition of the CN-PF pathway in this study. Even if we'd manage to combine these two technologies, we cannot certify that the recorded neurons would be the ones projecting to PF in this case. Moreover, even if we observed an inhibition of CN neurons, stimulations of PCs inhibiting CN neurons and chemogenetic inhibition of CN neurons would interfere in the interpretation of the exact mechanisms underlying this inhibition. However, Varani and

colleagues in our team showed that inhibitory DREADDs decreased by ~30% the firing rate of CN neurons (Varani *et al.*, 2020). Lines 389-390 were added in order to alleviate this concern.

Figure 7:

1. What animals were used for the analysis of Δ FosB? Were these the same animals used in Figure 1? If yes, why does the n per group differ from that presented in Figure 1? Please specify why and if animals were excluded from the analysis. If this is a new batch of animals, please report results for the TH analysis as supplemental data.

The animals used in the analysis of Δ FosB constitute a subset of the animals used in the main behavioral experiment (Figure1). This information has been added in the Results section (line 448) and in the Methods section (lines 824-826). No animals have been excluded from the study but the total of mice used of LID scoring were dispatched into several groups to perform the analysis of Δ FosB, the analysis of TH, and *ex vivo* experiments. This information has been added in the "Methods" section (lines 832-836). Some animals were excluded from the study for 1) lack of LID expression in the levodopa-treated condition (LID; N=2/14) and 2) for misplacing of the electrodes (all conditions; N=18/53).

2. Was the person performing Δ FosB cell counts blind to animal ID or experimental groups?

Yes. And several experimenters performed this analysis. It has been added in Methods, lines 824-826.

Reviewer #4 (Remarks to the Author):

This study demonstrates a remarkable influence of cerebellar neurostimulation on levodopa induced dyskinesia (LID) and striatal plasticity in a rodent model of Parkinson's disease. Daily stimulation via optogenetics of Purkinje cells in the orolingual region of the cerebellum evoked a surprisingly long-lasting reduction in oromotor LID with concomitant restoration of aberrant firing rates in motor cortex and thalamus, a reduction in LID induced striatal FOS expression, and a marked alteration of synaptic plasticity in striatal neurons that express D1 receptors. These findings are important; they reveal that cerebellar neurostimulation can provide a therapeutic means for improving dysfunctional physiological signaling across distributed brain circuits, most notably via the thalamic parafascicularis influence on dorsal striatum. The main weakness of the paper in its current form is that insufficient numbers of cerebellar nucleus neurons were recorded to make robust conclusions about population firing rates. The phrase "normalize brain activity" in the title overstates the findings, in which extracellular recordings subject to sampling biases were restricted to a subset of brain regions. Some of the figures are not easily interpretable. With appropriate revisions, this study provides a fundamental and exciting advance in our understanding of circuit mechanisms that

could be leveraged to ameliorate dyskinesia in Parkinson's and other disorders linked with striatal dysfunction.

Major

1. High variability in DCN firing rates across control conditions preclude any firm conclusions about restoration of normal firing. The sample sizes for DCN are in the tens or fewer, vs hundreds for motor cortex and PF. Many more units need to be recorded from DCN given heterogeneity of firing rates. The CV2 result looks interesting, but it's hard to interpret without seeing the corresponding firing patterns. Given challenges with recording stability in freely moving animals, were only largest neurons likely to be recorded? Results from motor cortex look more robust, but it's not clear whether recordings were restricted to pyramidal cells or also included interneurons.

This is indeed an important remark, thank you. Deep cerebellar nuclei are very heterogeneous structures with many different neuronal populations and a wide range of firing rate *in vivo* (0-100Hz) (Canto, Witter and Zeeuw, 2016). These features might explain the high variability observed between groups in the "Pre-Levodopa" phase and that's why we decided to make comparisons inside the groups themselves rather than between conditions as indeed the subsequent changes are difficult to interpret. Moreover, as we approached this question by recording the global activity of these structures, we do not exclude the fact that we may have recorded more than the principal cells of these structures (i.e. interneurons in both CN and M1) and therefore talked about global modulations. Canto and colleagues demonstrated a correlation between the soma size and the firing rate (Canto, Witter and Zeeuw, 2016), suggesting that we mostly recorded small neurons in CN. This has been reviewed and emphasized through the manuscript to avoid any misleading's (lines 232-235). Raw traces of both SHAM and LID animals were also added to Figure 1 to reinforce the observations made on cv2.isi. In the future, we would like indeed to be able to separate putative neuronal populations of these structures based on spike shapes' parameters.

We were able to add more new units for the DCN experiment.

In the motor cortex as well as in the other structures, the global activity of the area was recorded and analyzed without discriminating between the different cell types. Our analyses were done in order to the population responses to both levodopa treatment and cerebellar stimulation.

2. Citations about previous demonstrations of synaptic connections from DCN to parafasciculus and disynaptically to striatum are missing from the introduction and results and mentioned in passing in the Discussion (references 83-85). Figure 4 appears to be a simple replication of previous findings on DCN to PF projections in mice, with no new information that would justify inclusion in this paper.

Citations about CN projections to PF were added in the "Introduction" and "Results" sections. Moreover, the title and paragraph corresponding to this experiment have been fully revised in

order to better explain the rationale behind this experiment (Lines 350-375). The main point of this experiment, besides showing that these viruses worked in our hands for CN-PF projections, was to assess the level of collaterals from CN-PF projections in VAL. The synaptophysin experiment allowed us to make the statement that the behavioral effects on oral LID observed following chemogenetic inhibition of specific CN-PF projections were not due to inhibition of collaterals projecting to VAL. This information has been highlighted (lines 353-357 and 373-375) for better clarity on this concern.

Figures

1. General - figures with bar graphs have unexplained color dots which do not appear to correspond with the box plot values. The meaning of the box plots needs to be clarified in the figure legends. The many figures with multiple sequential box plots are difficult to parse - they seem cluttered with lines and dots but don't have an indication of the specific time window for each box plot.

The meaning of the boxplots, of the dots, and of the specific time window has been added in figures' legends (lines 1256-1258, 1290-1292, 1319-1321, 1374, 1418-1420, 1432-1435, 1455-1457, 1470-1472, 1485-1487, 1501-1503, 1553-1554, 1592-1594, 1630-1631, and 1669-1670). In addition, the corresponding figures have been uncluttered by the removal of the corrective group for better visualization of the most important findings. Boxplots represents the lower and the upper quartiles as well as the median of LID score. The vertical lines above and below the boxplot represent the median + the standard deviation (above) and the median - the standard deviation (below). The dots outside the boxplots represent the outliers of the distribution. This description could not be repeated for every panel in figure's legend due to characters' limitations but the full description has been added in the "Methods" section.

2. Figure 2. It looks like only a subset of PCs are labeled- is this an artifact of contrast adjustment?

That was indeed an artefact. We changed the panel showing ChR2-YFP expression in PCs in the cerebellar cortex of L7-ChR2-YFP mouse line (Figure 2b). Thank you. Moreover, it is important to note that the removal of the LED on CrusII can damage the cerebellar cortex underneath leading to less cerebellar cortex over CrusII on the right side, where the DCN were recorded.

2. Figure 5 - the mCherry and GFP signals are not adequately bright.

The concerned panels have been modified. Thank you.

3. Figure 7 - the FOS staining is not adequately bright.

The concerned panels have been modified. Thank you.

Minor

1. line 60 - please clarify the species here

Species of the corresponding cited studies have been specified (line 60).

2. line 63+ The motivation for the experiments needs to be made more explicit. "A first possibility" - for what?

Lines 62-64 were added to emphasize the need and motivations of the experiments performed in this study with the goal to shed light on the unknown mechanisms underlying LID alleviation. We hope this will be more explicit. Thank you.

3. line 73. relay to where? the fundamental circuit needs to be clarified here.

This information has been added (line 76). Moreover, a new figure showing a simplified version of the concerned circuitry, between the cerebellum, the thalamus, the basal ganglia, and the motor cortex, and the effects of cerebellar stimulations has been added for better clarity (Figure 8).

4. line 91. add references 84, 85 and note that PF is considered part of IL thalamus.

The information and corresponding references have been added (line 95). Thank you.

5. line 117. the classic model needs a reference

The pioneer references of the classical mouse model of LID have been added (Lundblad *et al.*, 2004; Francardo *et al.*, 2011) (line 123)

6. line 149 replace "none" with "no"

Changed. Thank you.

7. line 167, "were" - incorrect word

Changed. Thank you.

8. lines 302. remove "therefore"

Removed. Thank you.

9. line 575. provide specific parameters for theta stimulation.

Line 575 (now 709-712) has been rephrased for better clarity.

10. line 594 - missing s (s)pontaneous

Modified. Thank you.

11. lines 632-3. Add controls for off-target injections and verification of injection sites?

This is indeed an important concern. Controls for off-targets injections were not performed for this experiment. However, every injection site has been verified using confocal and animals that were not well injected were excluded from this experiment (N=1). This information has been added in the Methods Section lines 776-777. An inset in Figure 4a has been added as a representative example of the injection site in PF.

Reviewers' Comments:

Reviewer #1:

Remarks to the Author:

The authors have addressed my concerns. The clarity of the manuscript, and especially the figures, is greatly improved. This is a very interesting study.

Specific comments:

Line 61: '...their effect is mediated by the output cells of the cerebellar cortex, the Purkinje cells...'
– could the effect also be mediated by direct mossy fibre input on cerebellar nuclear cells, or intranuclear effects? I don't understand why cerebellar cortex is essential to this mechanism as observed in patients.

Line 87: plasticity's > plasticities

Line 98 (and many examples): PCs' > PC

Line 126,127: syntax 'oral LID'

I am a bit confused about control/sham/SHAM condition. Please standardise terminology. Lines 156-7 state that 'oral LID was more pronounced than axial and limb dyskinesia... .. compared to control mice'. If control mice have no dyskinesia, then this must be true by definition?

Line 171 – 'LID animals' > should this be 'unstimulated LID animals'?

Line 177 – 'No correlation was observed between the extent of the lesion and LID scores...': where or what is this data? Either include data or remove this statement.

Line 225 – '...no significant prevention of the decrease in global activity in DN and FN were observed...'. Earlier you state that levodopa does not increase FN activity, so remove subsequent reference to FN?

Line 317 – amount > amounts

Line 326 – This sentence is difficult to follow.

Line 346- onlyk

Reviewer #2:

Remarks to the Author:

The authors have submitted a substantially revised version of their manuscript. In this version they have nicely addressed each comment that I provided on the initial submission. The authors have therefore done an excellent job in addressing my concerns and have provided a detailed description of how they addressed each issue. This is an outstanding piece of work. I do not have any further concerns.

Reviewer #3:

Remarks to the Author:

This revised version of the manuscript reads much better than the first submission. My points of concerns have been sufficiently addressed. The topic is very interesting and timely.

Reviewer #4:

Remarks to the Author:

This is a good revision of a fascinating and important paper. I have no major criticisms.

Minor comments

1. The introduction starting with line 82 could be clarified beneficially for the reader if you were to start this paragraph with the companion phrase to the one in line 65 (A first hypothesis would be that...). eg, A second hypothesis is... Or, an alternative hypothesis is...

Given these are the two main hypotheses put forth, it would be best to address these directly in the Discussion, to clarify that they are not mutually exclusive.

2. Results line 1390140. "They did not exhibit severe dyskinesia of any kind" seems to imply that they exhibited mild dyskinesia. Please clarify.

3. line 174 remove ", and so"

4. line 457 replace "reaches" with "influences" or "affects". Also, it's worth noting that the mesodiencephalic junction also mediates cortical-cerebellar connections via inferior olive.

5. line 465: replace "evidence" with "demonstrate"

6. line 514: pluralize "pathway"

Figure legend 1. Define AIMS

Figure 4. A huge number of CN neurons are labelled- are these actually from restricted injections into PF, or might the injection site have also hit the mesodiencephalic junction (eg Darkschewitz, Interstitial nucleus of cajal) which is slightly caudal to the PF?

Figure 5D. Color figures need contrast enhancement - the red vs green vs blue are indistinguishable.

Figure 7. Color figures need contrast enhancement to visualize the labeled neurons

Figure 8 needs a legend. Also there are 4 types of lines - dotted vs not, thick vs thin - which make this a confusing figure. Not clear what "classical model" refers to.

Thank you for inviting us to revise our paper one last time to address the remaining concerns of our reviewers and your editorial requests. We are sending all the documents required.

Responses to Reviewers

Reviewer #1 (Remarks to the Author):

The authors have addressed my concerns. The clarity of the manuscript, and especially the figures, is greatly improved. This is a very interesting study.

Specific comments:

Line 61: “their effect is mediated by the output cells of the cerebellar cortex, the Purkinje cells”; could the effect also be mediated by direct mossy fibre input on cerebellar nuclear cells, or intra nuclear effects? I don’t understand why cerebellar cortex is essential to this mechanism as observed in patients.

This is an excellent remark. Their effect could indeed be directly modulated by mossy fibers on cerebellar nuclear cells or intra-nuclear effects. Based on their observations (metabolic changes in the cerebellar hemispheres and dentate nuclei) and other related observations in the motor cortex of dyskinetic patients, the authors suggested that their effect was mediated by the output cells of the cerebellar cortex, however, without a clear demonstration of this hypothesis. Lines 59-60 have been rephrased in order to better emphasize their hypothesis without, however, a clear understanding of the underlying mechanisms.

Line 87: plasticity’s; plasticities

Changed, thank you.

Line 98 (and many examples): PCs; PC

Changed, thank you.

Line 126,127: syntax “oral LID”

Changed, thank you.

I am a bit confused about control/sham/SHAM condition. Please standardize terminology. Lines 156-7 state that “oral LID was more pronounced than axial and limb dyskinesia; compared to control mice”. If control mice have no dyskinesia, then this must be true by definition?

It is indeed true by definition. We adapted the sentence for better clarity. Thank you.

SHAM/control/sham was indeed confusing. We used SHAM in capital letters in the entire text when speaking about non-dopaminergically depleted animals and modified the text accordingly. Other controls in other experiments are not SHAM. Thank you.

Line 171: “LID animals”; should this be “unstimulated LID animals”?

Indeed. This has been added.

Line 177 “No correlation was observed between the extent of the lesion and LID scores” where or what is this data? Either include data or remove this statement.

We removed the statement to clarify and unclutter the text. Thank you.

Line 225 “no significant prevention of the decrease in global activity in DN and FN were observed”. Earlier you state that levodopa does not increase FN activity, so remove subsequent reference to FN?

Indeed, that was confusing. We removed statements about the FN for preventive mice. Thank you.

Line 317: “amount” amounts

Changed. Thank you.

Line 326: This sentence is difficult to follow.

The sentence has been rephrased and simplified.

Line 346- onlyk

The extra “k” has been removed. Thank you.

Reviewer #2 (Remarks to the Author):

The authors have submitted a substantially revised version of their manuscript. In this version they have nicely addressed each comment that I provided on the initial submission. The authors have therefore done an excellent job in addressing my concerns and have provided a detailed description of how they addressed each issue. This is an outstanding piece of work. I do not have any further concerns.

Reviewer #3 (Remarks to the Author):

This revised version of the manuscript reads much better than the first submission. My points of concerns have been sufficiently addressed. The topic is very interesting and timely.

Reviewer #4 (Remarks to the Author):

This is a good revision of a fascinating and important paper. I have no major criticisms.

Minor comments

1. The introduction starting with line 82 could be clarified beneficially for the reader if you were to start this paragraph with the companion phrase to the one in line 65 (A first hypothesis would be that...). eg, A second hypothesis is... Or, an alternative hypothesis...

Given these are the two main hypotheses put forth, it would be best to address these directly in the Discussion, to clarify that they are not mutually exclusive.

Line 89 has been added for better clarity of the introduction's structure to the readers.

2. Results line 139-140. "They did not exhibit severe dyskinesia of any kind" seems to imply that they exhibited mild dyskinesia. Please clarify.

It implies that they exhibited only mild or no dyskinesia of any kind. The sentence was indeed unclear and has been rephrased.

3. line 174 remove " and so"

Removed. Thank you.

4. line 457 replace "reaches" with "influences" or "affects". Also, it's worth noting that the mesodiencephalic junction also mediates cortical-cerebellar connections via inferior olive.

Indeed. This has been changed.

5. line 465: replace "evidence" with "demonstrate"

Replaced. Thank you.

6. line 514: pluralize "pathway"

Pluralized. Thank you.

Figure legend 1. Define AIMS

This was indeed missing. The definition has been added. Thank you.

Figure 4. A huge number of CN neurons are labelled- are these actually from restricted injections into PF, or might the injection site have also hit the mesodiencephalic junction (eg Darkschewitz, Interstitial nucleus of cajal) which is slightly caudal to the PF?

That is an interesting point that should indeed be taken into account. However, in our neuroanatomical tracing experiment only mice in which the PF was correctly targeted were included. Only one mouse was removed for off-target injection (line 701-702).

Figure 5D. Color figures need contrast enhancement - the red vs green vs blue are indistinguishable.

The contrast has been enhanced. Thank you.

Figure 7. Color figures need contrast enhancement to visualize the labeled neurons

The contrast has been enhanced. Thank you.

Figure 8 needs a legend. Also there are 4 types of lines - dotted vs not, thick vs thin - which make this a confusing figure. Not clear what "classical model" refers to.

Thank you for seeing this oversight! The classical model refers to the classical rate model proposed by Albin et al. 1989 and Delong et al. 1990 to model the alterations occurring in the basal ganglia circuitry in both hypo- and hyperkinetic disorders. Figure 8 was adapted from Kishore et al. 2014 whose was adapted from Delong et al. 1990. This has been specified in the legend's figure. The specificity of dotted, plain, thick and thin lines were also specified in the legend's figure.